# The association between ozone and fine particles and mental health-related emergency department visits in California, 2005–2013

Angela-Maithy Nguyen[1,2], Brian J. Malig[1], Rupa Basu[1]*

**1** Air and Climate Epidemiology Section, California Office of Environmental Health Hazard Assessment, Oakland, California, United States of America, **2** Interdisciplinary Division, School of Public Health, University of California—Berkeley, Berkeley, California, United States of America

* rupa.basu@oehha.ca.gov

**Data Availability Statement:** As per the rules of the Office of Statewide Health Planning and Development's (OSHPD) CPHS, emergency department visit data cannot be shared due to its

## Abstract

Recent studies suggest that air pollutant exposure may increase the incidence of mental health conditions, however research is limited. We examined the association between ozone ($O_3$) and fine particles ($PM_{2.5}$) and emergency department (ED) visits related to mental health outcomes, including psychosis, neurosis, neurotic/stress, substance use, mood/affective, depression, bipolar, schizotypal/delusional, schizophrenia, self-harm/suicide, and homicide/inflicted injury, from 2005 through 2013 in California. Air monitoring data were provided by the U.S. EPA's Air Quality System Data Mart and ED data were provided by the California Office of Statewide Health Planning and Development. We used the time-series method with a quasi-Poisson regression, controlling for apparent temperature, day of the week, holidays, and seasonal/long-term trends. Per 10 parts per billion increase, we observed significant cumulative 7-day associations between $O_3$ and all mental health [0.64%, 95% confidence interval (CI): 0.21, 1.07], depression [1.87%, 95% CI: 0.62, 3.15], self-harm/suicide [1.43%, 95% CI: 0.35, 2.51], and bipolar [2.83%, 95% CI: 1.53, 4.15]. We observed 30-day lag associations between $O_3$ and neurotic disorder [1.22%, 95% CI: 0.48, 1.97] and homicide/inflicted injury [2.01%, 95% CI: 1.00, 3.02]. Same-day mean $PM_{2.5}$ was associated with a 0.42% [95% CI: 0.14, 0.70] increase in all mental health, 1.15% [95% CI: 0.62, 1.69] increase in homicide/inflicted injury, and a 0.57% [95% CI: 0.22, 0.92] increase in neurotic disorders per 10 µg/m³ increase. Other outcomes not listed here were not statistically significant for $O_3$ or $PM_{2.5}$. Risk varied by age group and was generally greater for females, Asians, and Hispanics. We also observed seasonal variation for outcomes including but not limited to depression, bipolar, schizophrenia, self-harm/suicide, and homicide/inflicted injury. Ambient $O_3$ or $PM_{2.5}$ may increase the risk of mental health illness, though underlying biological mechanisms remain poorly understood. Findings warrant further investigation to better understand the impacts of air pollutant exposure among vulnerable groups.

potentially identifying and sensitive information as defined by the Health Insurance Portability and Accountability Act. OSHPD data must be requested directly (https://oshpd.ca.gov/data-and-reports/research-data-request-information/) and is limited to researchers meeting their criteria for access to confidential data and requires institutional review board approval. The California CPHS can be contacted either by phone (916) 326-3660 or by email at cphs-mail@oshpd.ca.gov. The exposure data are publicly available from the U.S. Environmental Protection Agency's (EPA) Air Quality System Data Mart (https://aqs.epa.gov/aqsweb/documents/data_mart_welcome.html).

**Funding:** The author(s) received no specific funding for this work.

**Competing interests:** The authors have declared that no competing interests exist.

## Introduction

In the U.S., approximately 46.6 million adult Americans were living with at least one mental health condition in 2017, representing nearly 20 percent the population [1]. The risk of mental health disorders is associated with genetic predisposition and socioeconomic factors, while recent evidence has suggested the potential role of the environment as a risk factor [2].

Air pollutants have been associated with a range of acute and chronic physical health conditions such as lung cancer and cardiovascular and respiratory diseases [3–6]. There are also findings to suggest that ozone ($O_3$) and particulate matter with an aerodynamic diameter <2.5 μm ($PM_{2.5}$) in particular are linked to oxidative stress and neuroinflammation, which are pathways associated with mental health disorders [7, 8]. There is increasing interest to study mental health impacts due to air pollution, but existing research on the topic is limited [9]. Some evidence suggests that an association exists for a range of mental health conditions, including schizophrenia, depression, and self-harm/suicide, though these findings lack generalizability and findings are generally inconsistent thus far, since previous studies may differ by study design, region, exposure assessment, and types of mental health outcomes examined [10–15]. A review of the literature, including a recent systematic review, showed that most studies examining the relationship between fine particles and mental health, in general, were primarily based outside of the U.S. with most of the existing studies having been conducted in Canada, Western European countries, and Southeast Asian countries [9].

A previous study recently examined the association between carbon monoxide (CO) and nitrogen dioxide ($NO_2$) on mental health-related emergency department (ED) visits in California [16]. In this time-series study, we further expanded these findings to investigate the hypothesis that increased exposure to $O_3$ and $PM_{2.5}$ was associated with increased risk of ED visits related to mental health outcomes in California between 2005 and 2013. To consider potential seasonal trends and socioeconomic risk factors of several mental health outcomes, we also examined differences in associations by season and sub-groups defined by sex, age group, and race/ethnicity as well as regional characteristics related to community-level socioeconomic factors, obesity, and urbanicity as potential effect modifiers. In addition, we explored various lag times of exposure to $O_3$ and $PM_{2.5}$ on mental health-related ED visits.

## Materials and methods

### Outcome data

Statewide data on daily outpatient and inpatient ED visits were obtained from California's Office of Statewide Health Planning and Development (OSHPD) from January 1, 2005 through December 31, 2013. OSHPD also provided count data by age group, sex, and race/ethnicity. Our study included ED visits with primary diagnoses related to mental health-related conditions defined by the International Classification of Disease, 9th revision, Clinical Modification (ICD-9-CM) [17]. In addition to all mental health-related conditions (ICD-9-CM codes 290–319), we examined sub-categories of psychoses (290–299), neurotic (300–316), neurotic and stress-related (300; 308), mood and affective (296, 298, 309, 311), depression (311), bipolar (296), substance use-related (291, 292, 303–305), schizotypal/delusional-related (295, 297, 301), and schizophrenia (295) outcomes. We also examined external causes of injury related to self-harm or suicide ("self-harm/suicide"; E950-959) and homicide or injuries purposely inflicted by other persons ("homicide/inflicted injury"; E960-969).

### Environmental exposure variables

We obtained the daily 8-hour mean average concentrations of $O_3$ (in units of parts per billion, ppb), daily 24-hour mean average concentrations of $PM_{2.5}$ (μg/m$^3$), and daily 1-hour

mean average concentrations of $NO_2$ (ppb) monitored by the U.S. Environmental Protection Agency's (EPA) Air Quality System Data Mart [18]. Data on hourly mean air temperature and relative humidity were provided by the EPA's Air Quality System Data Mart, the National Oceanic and Atmospheric Administration and the California Irrigation Management Information System [19, 20]. We used this data to calculate daily mean apparent temperature, a heat index which incorporates temperature and relative humidity, as described previously [21]. We imputed missing values for $O_3$, $NO_2$, and apparent temperature using the R package 'mtsdi', which uses an imputation algorithm appropriate for multivariate normal time-series and accounts for both spatial and temporal correlation structures [22]. We did not impute values for $PM_{2.5}$ due to a larger percentage (> 5%) of missing values. For our $O_3$ analysis, we selected air monitoring stations which captured $O_3$ data for at least 90 percent of the observed days from December 1, 2004 through December 31, 2013 (3,117 days). In cases where there were multiple eligible monitors, the closest one was assigned. For our $PM_{2.5}$ analysis, we first selected air monitoring stations that captured data at least four days per month during the study period. We then selected air monitors that met this criterion for at least 90 percent of the total possible number of months during our study period. We then linked each $O_3$ and $PM_{2.5}$ monitor to nearby temperature and $NO_2$ monitors for confounding analyses.

## Statistical analysis

Air pollutant and daily temperature data were merged with ED visit data based on reported residential zip code, separately for each main exposure analysis. The same statistical methods were used for both $O_3$ and $PM_{2.5}$ analyses. To examine the association between these two air pollutants and several mental health outcomes, we conducted a two-stage time-series study using a quasi-Poisson generalized linear model. Air monitoring areas were captured as ZIP Code Tabulation Area (ZCTA) centroids located within a 20-kilometer radius of each $PM_{2.5}$ monitor and a 10-kilometer radius of each $O_3$ monitor. ZCTAs near multiple monitors were assigned to the closest one. We first obtained effect estimates for each air monitoring area, followed by a random effects meta-analysis to produce overall estimates for California and 95 percent confidence intervals. Our meta-analysis excluded monitors that did not include converge in our first stage of analysis. Each model, for each pollutant and all mental health outcomes, was adjusted for the following confounders: mean apparent temperature (same-day lag), day of the week, national holidays, and seasonal/long-term trends. Day of the week was coded as weekend versus weekday (dichotomous, 0/1). We used same-day mean apparent temperature because same-day or short-term mean temperature are more likely to be correlated with short-term pollution, thus more likely to confound the pollution effects. Results from a previous study on ambient temperature and mental health showed that stronger effects were observed for short-term temperatures [23]. National holidays included New Year's Day, Memorial Day, Independence Day, Labor Day, Thanksgiving, and Christmas. We combined Christmas, Memorial Day, and Thanksgiving as one dichotomous variable in our analytical models because the magnitude of effects between these holidays and all mental health conditions showed negative associations. New Year's Day, Independence Day, and Labor Day were represented as three dichotomous variables. Seasonal/long-term trends were adjusted for by using a natural cubic spline measured with 2 degrees of freedom (df) per year. We compared controlling for seasonal/long-term trends using 2 df versus 3 and 4 df and found that the quasi-Akaike's Information Criterion (qAIC) in our models worsened for both 3 and 4 df.

For our $O_3$ analyses, we examined short-term single-day lags of the same day up to 7 days, in addition to cumulative weekly (lag 0–6 days) and longer-term monthly (lag 0–29 days). For our $PM_{2.5}$ analyses, we examined same-day lags up to 7 days and cumulative 1-day lags. $PM_{2.5}$

data are typically collected every third or sixth day, thus we created a cumulative 1-day lag with at least 50% of data available in each cumulative lag. We did not explore $PM_{2.5}$ cumulative lags beyond 1 day due to the limitations of the data. $O_3$ models were compared to one another separately for each outcome to select the best-fitting lag fit based on each model's sum of qAICs and all $PM_{2.5}$ models were compared via statistical significance. We also accounted for potential confounding from co-pollutants by repeating our analyses for all outcomes in separate two pollutant models, including $NO_2$, $PM_{2.5}$, or $O_3$. Statistical significance was determined at the p <0.05 level.

## Secondary analyses

In our secondary analyses, we examined for potential seasonal effect modification by stratifying data by warm (May to October) and colder months (November to April) for all mental health outcomes considered. We also evaluated potential effect modification by several categorical variables: sex (male or female), age group (0–18, 19–34, 35–49, 50–64, 65 years and older), and race/ethnicity (non-Hispanic White, non-Hispanic Black, non-Hispanic Asian, and Hispanic). We stratified analyses by each of these variables and assessed for any significant differences between categories. In addition, we explored heterogeneity by percent obesity [low (<29%) versus high (≥29%) percent of obesity] and several socioeconomic-related variables including ZCTA-level percent below the federal poverty threshold [low (<15%) versus high (≥15%) percent of those below the federal poverty line], median household income [low (< $50,000) versus high (≥$50,000) median household income], percent unemployment [low (<8%) versus high (≥8%) percent of unemployment], and percent with a Bachelor's level of education or higher [low (<30%) versus high (≥30%) percent of college educational attainment or higher]. We then generated the percent obesity and socioeconomic-related variables using the ZCTA centroid population-weighted averages for each air monitor in our study population and each variable was dichotomized using the highest quartile as the cut-off point. Percent obese was created using 2013–2017 data from the Uniform Data System Mapper, a federal health programming geographic tracking tool [24]. Percent with a Bachelor's level of education or higher, percent below the poverty threshold, and median household income were derived from the 2011 American Community Survey [25]. Lastly, we created the variable urbanicity, defined as rural (<500 individuals per square mile) versus urban (≥500 individuals per square mile), as a binary variable using the 2010 U.S. census data on population density (population per square mile). For all effect modification analyses, we conducted tests of interaction to evaluate for statistically significant differences across strata estimates [26]. We applied the same statistical methods in our secondary analyses as conducted in our main analyses.

In addition to single-day and average cumulative-day lags, we examined delayed effects for $O_3$ exposure using a distributed lag modeling framework. This framework is used to describe linear, non-linear, and delayed effects between the primary exposure and the outcome of interest [27]. In addition to a natural cubic spline as used in our main models, our distributed lag models included knots equally spaced on the log scale. The purpose of setting knots on the log scale is to parametrize the exposure and lag-response relationship. We modeled the $O_3$ dose-response linearly and the lag-response as a natural spline with knots equally spaced on the log scale, resulting in 3 knots for 7-day lag periods and 6 for 30-day periods. We then compared our distributed lag models to our single-day and cumulative-lag models to assess model fit using the qAIC. Our distributed lag modeling was conducted using the R package 'dlnm' [28]. We were not able to conduct a distributed lag analysis for our $PM_{2.5}$ data due to the proportion of missing values across air monitors.

For each mental health outcome of interest, we reported main findings as percent change in risk of ED visits per 10 ppb increase in $O_3$ and 10 μg/m3 increase in $PM_{2.5}$. All Geographic

Information Systems (GIS) tasks were done using ArcMap, version 10.7.1 and all statistical analyses were conducted using R, version 3.6.1 [29, 30]. Institutional Review Board approval was acquired by the Office of Environmental Health Hazard Assessment from California's Committee for the Protection of Human Subjects prior to beginning this study.

## Results

### Descriptive findings

Our selection and exclusion criteria for the $O_3$ exposure analysis resulted in 91 air monitors. Fig 1 shows the locations of the study population's $O_3$ air monitors in California. Table 1 describes the study population encompassed by these monitors for all considered mental health outcomes and by sex, race/ethnicity, and age group. During our study period, there were a total of 1,997,992 cases of all mental-health related ED visits. Among these cases, 99,679 ED visits were categorized as schizophrenia, 159,801 as depression, and 158,906 as bipolar-related. Additionally, there were 193,530 visits due to self-harm/suicide, and 632,787 visits due to homicide or injuries purposely inflicted by other persons. There were differences by sex, race/ethnicity, and age group across all mental health-related visits and sub-categories of interest. Males made up more than 50 percent of ED visits for all mental health visits and most of the sub-categories, while females made up more than 50 percent of ED visits related to

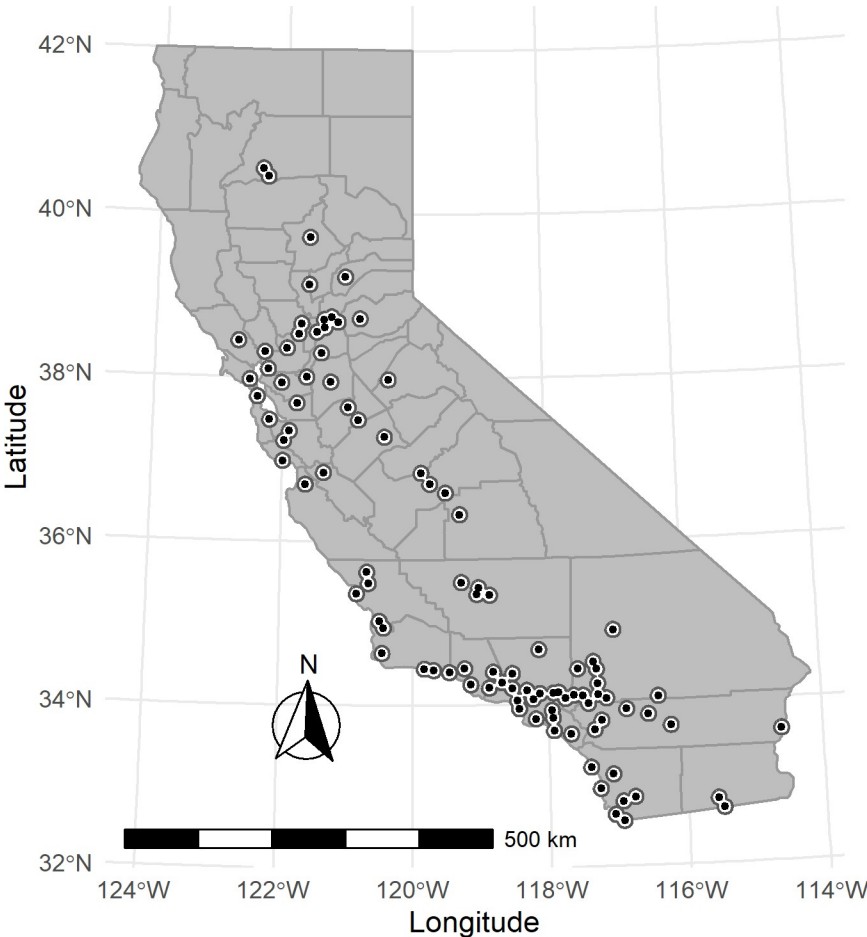

**Fig 1. Map of study $O_3$ air monitors (with 10-kilometer buffer) in California (N = 91).**

**Table 1. $O_3$ and $PM_{2.5}$ exposure study population characteristics: All mental health-related emergency department visits and sub-categories from 2005 through 2013 in California (N = 91 $O_3$ and 51 $PM_{2.5}$ air monitors).**

| | All mental health-related[1] | | Psychosis[2] | | Neurotic disorders[3] | | Neurotic/stress[4] | | Substance use[5] | | Mood/affective[6] | | Depression[7] | | Bipolar[8] | | Schizotypal/delusional[9] | | Schizophrenia[10] | | Self-Harm/Suicide[11] | | Homicide/Inflicted Injury[12] | |
|---|---|---|---|---|---|---|---|---|---|---|---|---|---|---|---|---|---|---|---|---|---|---|---|---|
| | $O_3$ | $PM_{2.5}$ | $O_3$ | $PM_{2.5}$ | $O_3$ | $PM_{2.5}$ | $O_3$ | $PM_{2.5}$ | $O_3$ | $PM_{2.5}$ | $O_3$ | $PM_{2.5}$ | $O_3$ | $PM_{2.5}$ | $O_3$ | $PM_{2.5}$ | $O_3$ | $PM_{2.5}$ | $O_3$ | $PM_{2.5}$ | $O_3$ | $PM_{2.5}$ | $O_3$ | $PM_{2.5}$ |
| **Total N** (%) | 1997992 (100) | 1174394 (100) | 620048 (31) | 370508 (32) | 1376774 (69) | 802751 (68) | 527024 (26) | 308703 (26) | 683674 (34) | 387605 (33) | 493744 (25) | 304263 (26) | 159801 (8) | 95768 (8) | 158906 (8) | 96466 (8) | 117129 (6) | 68996 (6) | 99679 (5) | 58568 (5) | 193530 (10) | 109399 (9) | 632787 (32) | 390339 (33) |
| **Sex** | | | | | | | | | | | | | | | | | | | | | | | | |
| Males | 53 | 53 | 59 | 59 | 51 | 50 | 39 | 38 | 67 | 68 | 51 | 50 | 46 | 46 | 49 | 48 | 64 | 64 | 66 | 66 | 41 | 41 | 69 | 69 |
| Females | 47 | 47 | 41 | 41 | 49 | 50 | 61 | 62 | 33 | 32 | 49 | 49 | 54 | 54 | 51 | 52 | 36 | 36 | 34 | 34 | 59 | 59 | 31 | 31 |
| **Age group** | | | | | | | | | | | | | | | | | | | | | | | | |
| 0–18 | 11 | 11 | 6 | 6 | 13 | 13 | 9 | 9 | 10 | 10 | 12 | 12 | 15 | 16 | 12 | 13 | 3 | 4 | 2 | 2 | 21 | 22 | 18 | 19 |
| 19–34 | 32 | 32 | 30 | 30 | 33 | 33 | 36 | 37 | 29 | 29 | 33 | 34 | 31 | 31 | 33 | 33 | 33 | 33 | 34 | 34 | 37 | 38 | 47 | 47 |
| 35–49 | 30 | 30 | 32 | 32 | 29 | 29 | 29 | 29 | 33 | 33 | 30 | 30 | 29 | 29 | 31 | 31 | 35 | 35 | 37 | 37 | 25 | 25 | 23 | 22 |
| 50–64 | 19 | 19 | 21 | 21 | 19 | 18 | 17 | 16 | 23 | 23 | 18 | 18 | 18 | 18 | 19 | 18 | 23 | 22 | 24 | 23 | 13 | 12 | 10 | 10 |
| 65+ | 8 | 8 | 1 | 11 | 7 | 7 | 9 | 9 | 5 | 5 | 7 | 7 | 6 | 5 | 5 | 5 | 5 | 5 | 4 | 4 | 3 | 3 | 2 | 2 |
| **Race/ethnicity[13]** | | | | | | | | | | | | | | | | | | | | | | | | |
| White | 48 | 44 | 52 | 47 | 47 | 42 | 44 | 39 | 54 | 50 | 48 | 43 | 51 | 47 | 52 | 46 | 45 | 41 | 43 | 39 | 54 | 50 | 34 | 28 |
| Black | 10 | 12 | 13 | 16 | 9 | 10 | 8 | 9 | 8 | 10 | 13 | 17 | 11 | 13 | 13 | 16 | 20 | 23 | 21 | 24 | 8 | 9 | 16 | 20 |
| Asian | 3 | 3 | 4 | 4 | 3 | 3 | 4 | 4 | 2 | 2 | 4 | 4 | 3 | 3 | 3 | 4 | 5 | 5 | 5 | 5 | 4 | 4 | 3 | 3 |
| Hispanic | 30 | 32 | 23 | 25 | 33 | 36 | 37 | 40 | 29 | 32 | 25 | 27 | 25 | 27 | 23 | 26 | 22 | 23 | 22 | 23 | 26 | 28 | 38 | 40 |

Outcome categories were defined by the following ICD-9 codes

[1] 290–319

[2] 290–299

[3] 300–316

[4] 300, 308

[5] 291, 292, 303–305

[6] 296, 298, 309, 311

[7] 311

[8] 296

[9] 295, 297, 301

[10] 295

[11] E950–959

[12] E960–969

[13] White, Black, and Asian categories are of non-Hispanic origin.

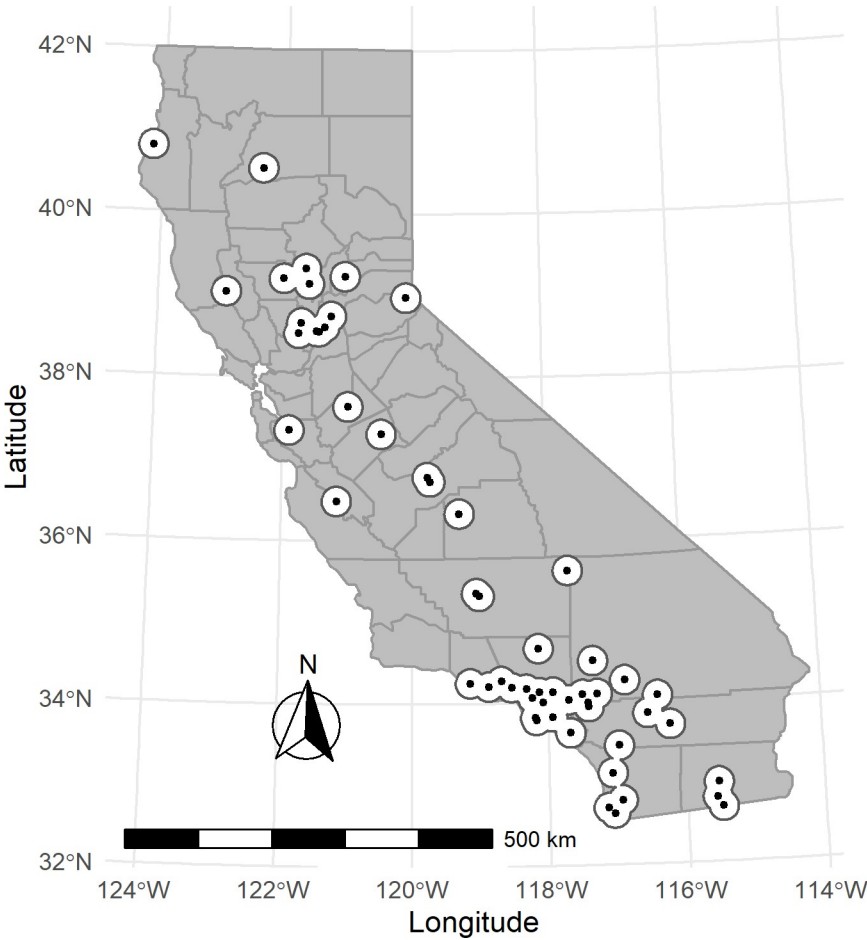

**Fig 2. Map of study PM$_{2.5}$ air monitors (with 20-kilometer buffer) in California (N = 51).**

neurotic and stress-related, depression, self-harm/suicide, and bipolar disorders. Nearly 50 percent of all mental health disorder ED visits were non-Hispanic White (White), while Hispanics made up 30 percent. Non-Hispanic Blacks (Blacks; 10%) and non-Hispanic Asians (Asians; 3%) had smaller proportions of all mental health disorder ED visits. Those between 19–34 years and 35–49 years made up 32 percent and 30 percent of all mental health visits, respectively, while 50–64 year old age group constituted 19 percent, children 0–18 year ages made up 11 percent, and those aged 65 years and older made up 8 percent for all mental health ED visits. These proportions were consistent across our categories of mental health conditions, except for homicide/inflicted injury and schizophrenia. For the category of homicide/inflicted injury, nearly 47 percent of ED visits were in the age group 19 to 34 years and 37 percent of schizophrenia-related visits were 35 to 49 years. Our PM$_{2.5}$ exposure analysis included 51 air monitors across California. Fig 2 shows the locations of the study population's PM$_{2.5}$ air monitors in California. This analysis consisted of 1,174,394 cases for all mental health-related ED visits from 2005 to 2013. The distribution of sub-categories was similar to our O$_3$ study population (Table 1).

Table 2 summarizes the daily mean and median distributions of O$_3$ and PM$_{2.5}$ for all air monitors during the study period. The overall daily mean concentration of O$_3$ was 30 ppb with a slightly higher daily mean concentration during warmer months (36 ppb) and lower

**Table 2. Mean daily O$_3$ (ppb) and PM$_{2.5}$ (μg/m$^3$) exposure by season, California, 2005–2013.**

|  | Mean | | SD | | Lower 5th Percentile | | Median | | Upper 95th Percentile | |
| --- | --- | --- | --- | --- | --- | --- | --- | --- | --- | --- |
|  | O$_3$ | PM$_{2.5}$ | O$_3$ | PM$_{2.5}$ | O$_3$ | PM$_{2.5}$ | O$_3$ | PM$_{2.5}$ | O$_3$ | PM$_{2.5}$ |
| **Overall** | 30 | 12 | 14 | 9 | 9 | 3 | 30 | 10 | 54 | 29 |
| **Warm** | 36 | 11 | 13 | 7 | 18 | 4 | 34 | 11 | 60 | 22 |
| **Cold** | 24 | 13 | 12 | 11 | 7 | 2 | 24 | 10 | 44 | 35 |

SD: Standard Deviation; ppb: parts per billion; Warm: (May-October), Cold: (November-April)

concentrations in the colder months (24 ppb). The overall daily mean and median distributions for the study sample PM$_{2.5}$ air monitors were 12 μg/m$^3$ and 10 μg/m$^3$, respectively. The mean concentration of PM$_{2.5}$ was higher in the colder months (13 μg/m$^3$) compared to warmer months (11 μg/m$^3$).

## Main findings

The results for O$_3$ effects at lag 0, lag 1, lag 2, lag 7, lag 0–6, and lag 0–29 are shown in Table 3. Based on the qAICs, we found that our cumulative 7-day models consistently had the best fit for all mental health, neurotic and stress, mood and affective disorders, depression, schizophrenia, schizotypal/delusional, self-harm/suicide, and bipolar disorders and that our cumulative 30-day lag models had the best fit for neurotic, homicide/inflicted injury, psychotic, and substance use-related disorders for O$_3$. Fig 3 shows the results from the cumulative 7-day and 30-day lag models for all mental health outcomes of interest. For mood and affective disorders, the 7-day average of the daily mean level of O$_3$ (per 10 ppb increase) exposure was associated with a 1.52% [95% confidence interval (CI): 0.69, 2.36] increase in ED visits. Within this broad category, we observed increased risks for bipolar [2.83%, 95% CI: 1.53, 4.07] and depression [1.86%, 95% CI: 0.62, 3.15] outcomes. Additionally, O$_3$ was associated with a 1.43% [95% CI%: 0.35, 2.51] increase for self-harm/suicide visits. The 30-day average of the daily mean level of O$_3$ was associated with a 2.01% [95% CI: 1.00, 3.02] increase in homicide/inflicted injury visits and a 1.22% [95% CI: 0.48, 1.97] increase in neurotic disorders (Fig 3). We found no evidence of NO$_2$ or PM$_{2.5}$ confounding in any of our two-pollutant models (results not shown).

The results for PM$_{2.5}$ at lag 0, lag 1, lag 2, and lag 7 are shown in Table 3. In our PM$_{2.5}$ exposure analyses, we found statistically significant associations for same-day (lag 0) models. Short-term exposure to PM$_{2.5}$ (per 10 μg/m$^3$ increase) was associated with all mental health outcomes [0.42%, 95% CI: 0.14, 0.70], neurotic disorder [0.57%, 95% CI: 0.22, 0.92], and homicide/inflicted injury outcomes [1.15%, 95% CI: 0.61, 1.69]. Fig 4 shows the results from both same-day models for all examined outcomes. We did not find any significantly increased risks for other mental health outcomes examined, though results for neurotic and stress, mood and affective disorders, and substance use were suggestive of associations (*p = 0.06*) (Fig 4). After controlling for NO$_2$ and O$_3$, we did not find any evidence of confounding in our two-pollutant models (results not shown).

## Sub-group analyses: Sex, age, and race/ethnicity

We observed differences in percent changes of risk by sex across different mental health outcomes following O$_3$ and PM$_{2.5}$ exposure (Figs 5–7). After testing for interaction, we found that females had a significantly increased risk for substance use disorder [1.77%, 95% CI: 0.60, 2.96] compared to males [-0.01%, 95% CI: -1.20, 1.19 (*p$_{diff}$ = 0.04*)] following O$_3$ exposure (Fig 5A). While we did not observe PM$_{2.5}$ and schizotypal/delusional-related ED visits in our main

**Table 3. Percentage change (95% CI) in the risk of all mental health-related ED visits for $O_3$ (per 10 ppb increase) and $PM_{2.5}$ (per 10 μg/m³ increase) using different lag days in single pollutant models.**

| | Lag 0 | | Lag 1 | | Lag 2 | | Lag 7 | | Lag 0–6 | Lag 0–29 |
|---|---|---|---|---|---|---|---|---|---|---|
| Outcome | $O_3$ | $PM_{2.5}$ | $O_3$ | $PM_{2.5}$ | $O_3$ | $PM_{2.5}$ | $O_3$ | $PM_{2.5}$ | $O_3$ | $O_3$ |
| All mental health-related[1] | -0.31 (-0.58, -0.03) | 0.42 (0.14, 0.70) | 0.19 (-0.06, 0.43) | -0.03 (-0.42, 0.37) | 0.21 (-0.05, 0.46) | 0.04 (-0.27, 0.35) | 0.16 (-0.08, 0.41) | -0.29 (-0.56, -0.01) | 0.64 (0.21, 1.07) | 0.92 (0.29, 1.55) |
| Psychosis[2] | -0.42 (-0.85, 0.02) | 0.10 (-0.37, 0.56) | -0.04 (-0.46, 0.37) | 0.01 (-0.56, 0.59) | 0.23 (-0.16, 0.62) | 0.06 (-0.40, 0.53) | -0.14 (-0.58, 0.30) | -0.22 (-0.82, 0.38) | 0.53 (-0.15, 1.21) | 0.39 (-0.49, 1.27) |
| Neurotic[3] | -0.25 (-0.57, 0.07) | 0.57 (0.22, 0.92) | 0.29 (-0.01, 0.58) | -0.01 (-0.50, 0.48) | 0.20 (-0.10, 0.49) | 0.05 (-0.33, 0.44) | 0.31 (0.04, 0.58) | -0.29 (-0.60, 0.03) | 0.72 (0.22, 1.22) | 1.22 (0.48, 1.97) |
| Neurotic/Stress[4] | -0.10 (-0.49, 0.29) | 0.56 (-0.03, 1.16) | 0.57 (0.18, 0.96) | 0.22 (-0.58, 1.03) | 0.49 (0.06, 0.93) | 0.27 (-0.30, 0.86) | 0.11 (-0.30, 0.52) | -0.21 (-0.71, 0.28) | 0.64 (-0.03, 1.30) | 0.53 (-0.33, 1.40) |
| Substance use[5] | -0.36 (-0.79, 0.06) | 0.44 (-0.02, 0.90) | 0.00 (-0.36, 0.36) | 0.03 (-0.43, 0.49) | -0.26 (-0.68, 0.16) | 0.10 (-0.36, 0.56) | 0.51 (0.12, 0.90) | -0.10 (-0.75, 0.55) | -0.03 (-0.60, 0.55) | 0.61 (-0.35, 1.58) |
| Mood/Affective[6] | -0.05 (-0.48, 0.39) | 0.46 (-0.06, 0.98) | 0.34 (-0.05, 0.73) | -0.07 (-0.77, 0.64) | 0.63 (0.18, 1.08) | 0.02 (-0.49, 0.53) | -0.21 (-0.61, 0.18) | -0.32 (-0.86, 0.22) | 1.52 (0.69, 2.36) | 1.53 (0.42, 2.64) |
| Depression[7] | -0.12 (-0.82, 0.59) | 0.58 (-0.40, 1.57) | 0.16 (-0.66, 0.99) | -0.57 (-1.89, 0.77) | 0.33 (-0.55, 1.21) | -0.30 (-1.20, 0.60) | -0.34 (-1.01, 0.34) | -0.49 (-1.52, 0.55) | 1.87 (0.62, 3.15) | 2.72 (0.90, 4.58) |
| Bipolar[8] | 0.06 (-0.66, 0.78) | -0.10 (-1.18, 0.99) | 0.51 (-0.17, 1.19) | 0.40 (-0.65, 1.45) | 0.77 (0.09, 1.46) | -0.21 (-1.11, 0.70) | 0.003 (-0.68, 0.69) | 0.24 (-0.64, 1.13) | 2.83 (1.53, 4.15) | 2.81 (1.32, 4.33) |
| Schizotypal/Delusional[9] | -0.81 (-1.64, 0.03) | -0.17 (-1.21, 0.87) | -0.35 (-1.27, 0.57) | -0.20 (-1.24, 0.84) | -0.42 (-1.47, 0.63) | 0.52 (-0.79, 1.85) | -0.06 (-0.86, 0.75) | 0.13 (-1.17, 1.45) | -0.08 (-1.51, 1.36) | -0.42 (-2.31, 1.50) |
| Schizophrenia[10] | -0.79 (-1.70, 0.13) | -0.37 (-1.49, 0.76) | 0.00 (-0.91, 0.92) | 0.01 (-1.29, 1.33) | -0.42 (-1.29, 0.45) | 0.50 (-0.70, 1.72) | 0.09 (-0.78, 0.97) | 0.39 (-1.16, 1.95) | 0.31 (-1.04, 1.67) | 0.24 (-1.63, 2.14) |
| Self-Harm/Suicide[11] | 0.48 (-0.17, 1.14) | -0.43 (-1.23, 0.37) | 1.86 (1.21, 2.51) | 0.16 (-0.62, 0.96) | 1.18 (0.58, 1.79) | -0.13 (-1.01, 0.76) | 0.04 (-0.57, 0.64) | -1.19 (-1.97, -0.40) | 1.43 (0.35, 2.51) | 1.65 (0.36, 2.95) |
| Homicide/Inflicted Injury[12] | 0.23 (-0.14, 0.60) | 1.15 (0.62, 1.69) | 0.90 (0.46, 1.34) | 0.71 (0.26, 1.16) | 0.44 (0.09, 0.80) | 0.69 (0.18, 1.19) | 0.53 (0.09, 0.97) | -0.68 (-1.25, -0.12) | 1.08 (0.42, 1.74) | 2.01 (1.00, 3.02) |

Outcome categories were defined by the following ICD-9 codes:

[1] 290–319

[2] 290–299

[3] 300–316

[4] 300, 308

[5] 291, 292, 303–305

[6] 296, 298, 309, 311

[7] 311

[8] 296

[9] 295, 297, 301

[10] 295

[11] E950-959

[12] E960-969

CI: Confidence Interval

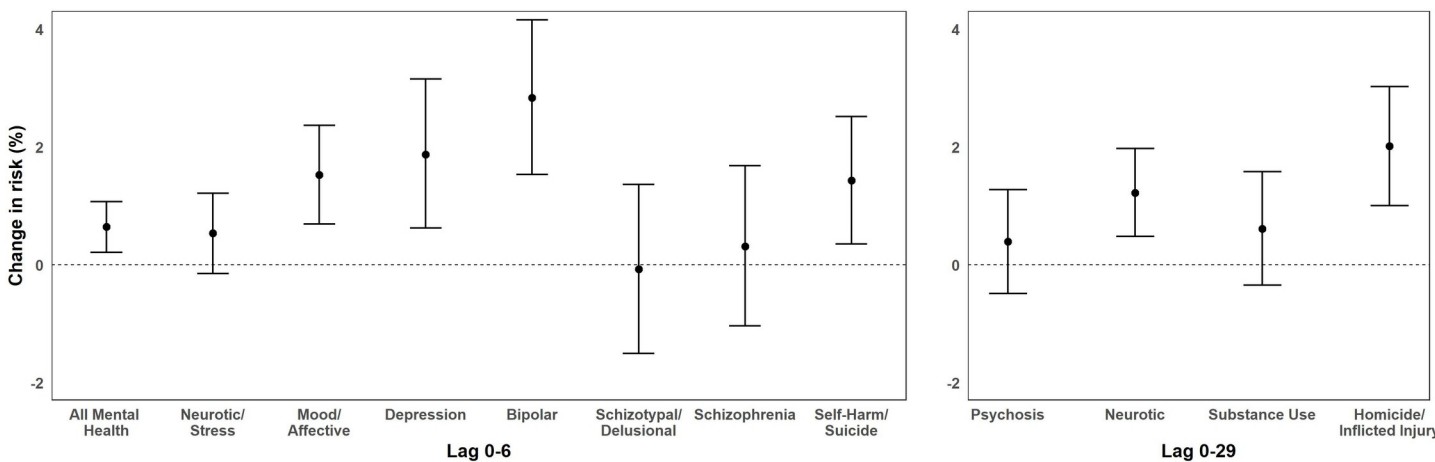

**Fig 3. Cumulative 7-day and 30-day lag effects for $O_3$ (per 10 ppb increase) and all mental health outcomes from 2005 through 2013 in California.** Bars: 95% Confidence Interval; ppb: parts per billion.

analyses, we found suggestive evidence of interaction by sex for schizotypal/delusional-related ED visits. We observed increased risks for schizotypal/delusional-related ED visits among females [1.34%, 95% CI: -0.54, 3.25] compared to males [-0.80%, 95% CI: -2.09, 0.51; ($p_{diff}$ = 0.07)] (Fig 5B).

Compared to age group 19 to 34 years, $O_3$ exposure among age group 0 to 18 years was significantly associated with increased risks for all sub-categories of mental health outcomes ($p_{diff}$ = <0.01) except for neurotic and stress disorders, schizophrenia and schizotypal/delusional-related disorders. Three of the highest risk categories for the 0 to 18 year age group were ED visits related to psychosis [12.71%, 95% CI: 8.82, 16.73], bipolar disorder [11.56%, 95% CI: 7.39, 15.90] and depression [9.37%, 95% CI: 5.97, 12.87] (Fig 6A). Mood and affective disorders were not included in this figure since we included findings for depression and bipolar

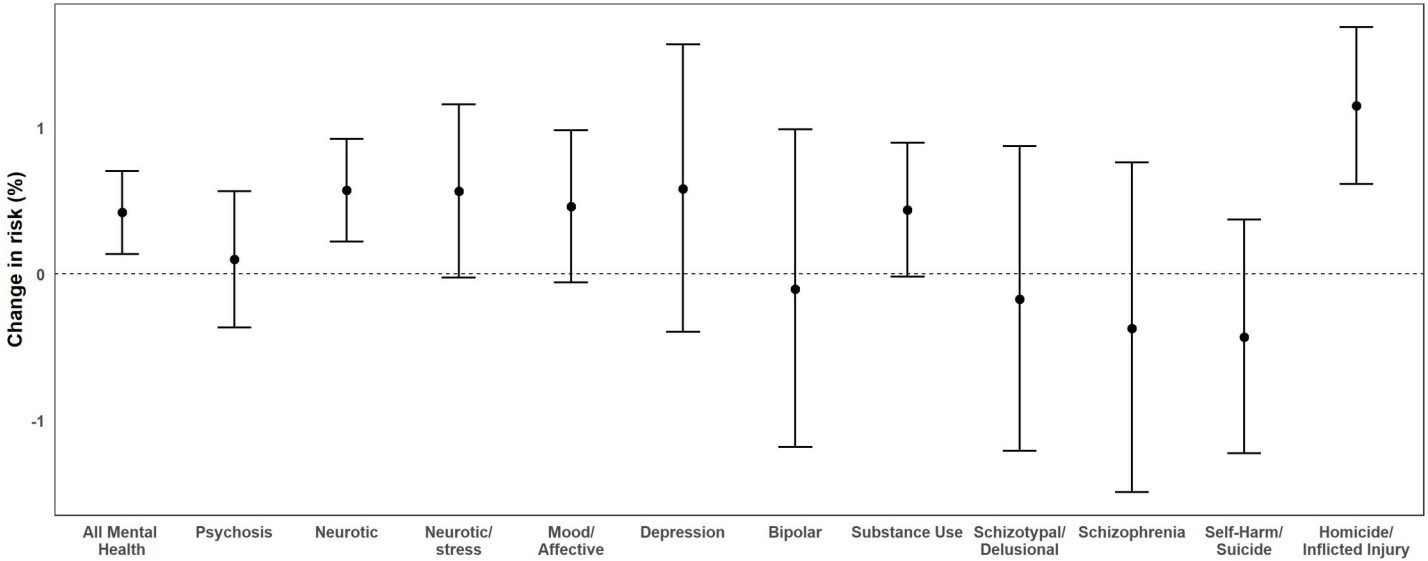

**Fig 4. Same-day (lag 0) lag effects for $PM_{2.5}$ (per 10 μg/m$^3$ increase) and all mental health outcomes from 2005 through 2013 in California.** Bars: 95% Confidence Interval.

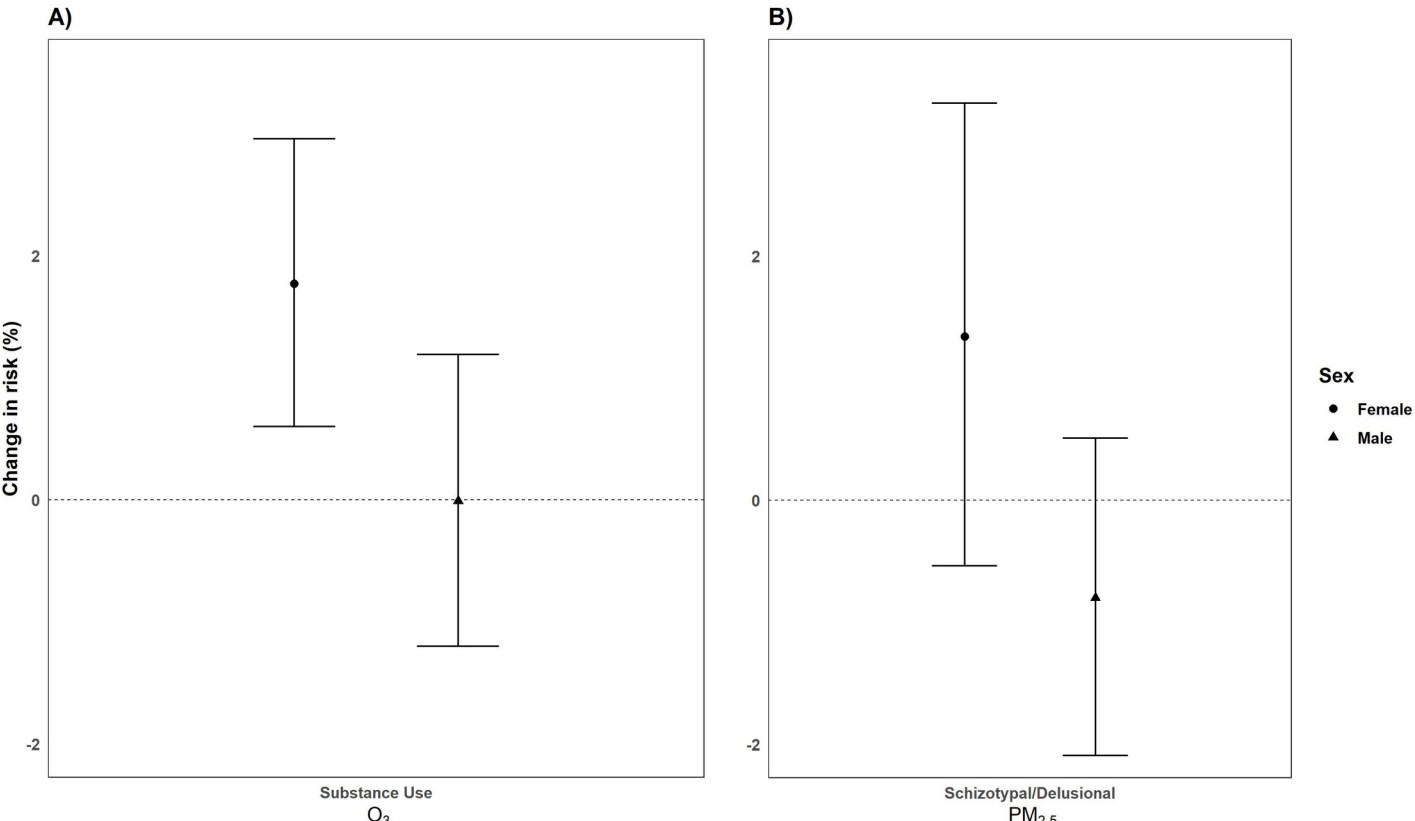

**Fig 5.** Percentage change (95% CI) in the risk of ED visits for $O_3$ (per 10 ppb increase) (A) and $PM_{2.5}$ (per 10μg/m³ increase) (B) by gender from 2005–2013 in California.[1] Bars: 95% Confidence Interval; CI: Confidence Interval; ED: Emergency Department; ppb: parts per billion. [1]Only significant outcomes shown.

disorder, two specific conditions within the broader category. In contrast, $PM_{2.5}$ had significantly increased mental health risks for the older age groups and these associations were only observed for all mental health, neurotic and stress, and schizotypal/delusional disorders (Fig 6B). Compared to the age group 19 to 34 years [-0.14%, 95% CI: -0.96, 0.68], age groups 35 to 49 years had significantly increased risks for neurotic and stress [1.47%, 95% CI: 0.41, 2.54 ($p_{diff}$ = 0.02)]. There were also increased risks for schizotypal/delusional disorders [1.06%, 95% CI: -0.65, 2.80] for age groups 35 to 49 years compared to age group 19 to 34 years [-1.56%, 95% CI: -3.33, 0.23 ($p_{diff}$ = 0.04)]. In addition, those 65 years and older were at significantly increased risks for all mental health [1.39%, 95% CI: 0.49, 2.29], neurotic and stress [2.0%, 95% CI: 0.19, 3.83], and schizotypal/delusional disorders [5.3%, 95% CI: 0.43, 10.51] compared to the age group 19 to 34 years [all mental health: 0.26, 95% CI: -0.26, 0.79 ($p_{diff}$ = 0.03); neurotic and stress: -0.14%, 95% CI: -0.96, 0.68 ($p_{diff}$ = 0.04); schizotypal/delusional: -1.56%, 95% CI: -3.33, 0.23 ($p_{diff}$ = 0.01)].

While we observed associations for all race/ethnic groups in our $O_3$ analysis, there were significantly increased risks observed for Hispanics compared to Whites for depression [0.98%, 95% CI: -0.60, 2.59] and homicide/inflicted injury-related [1.13%, 95% CI: -0.13, 2.39] ED visits (Fig 7A). Exposure to $O_3$ was significantly associated with a 3.78% increase [95% CI: 1.69, 5.91 ($p_{diff}$ = 0.04)] in depression and a 3.13% increase [95% CI: 1.77, 4.51 ($p_{diff}$ = 0.03)] in homicide/inflicted injury outcomes among Hispanics. We did not observe any significant differences in risk among Blacks or Asians, though Blacks were at greatest risk for psychosis [2.40%, 95% CI: -0.16, 5.04] and Asians were at greatest risk for substance use [9.20%, 95% CI:

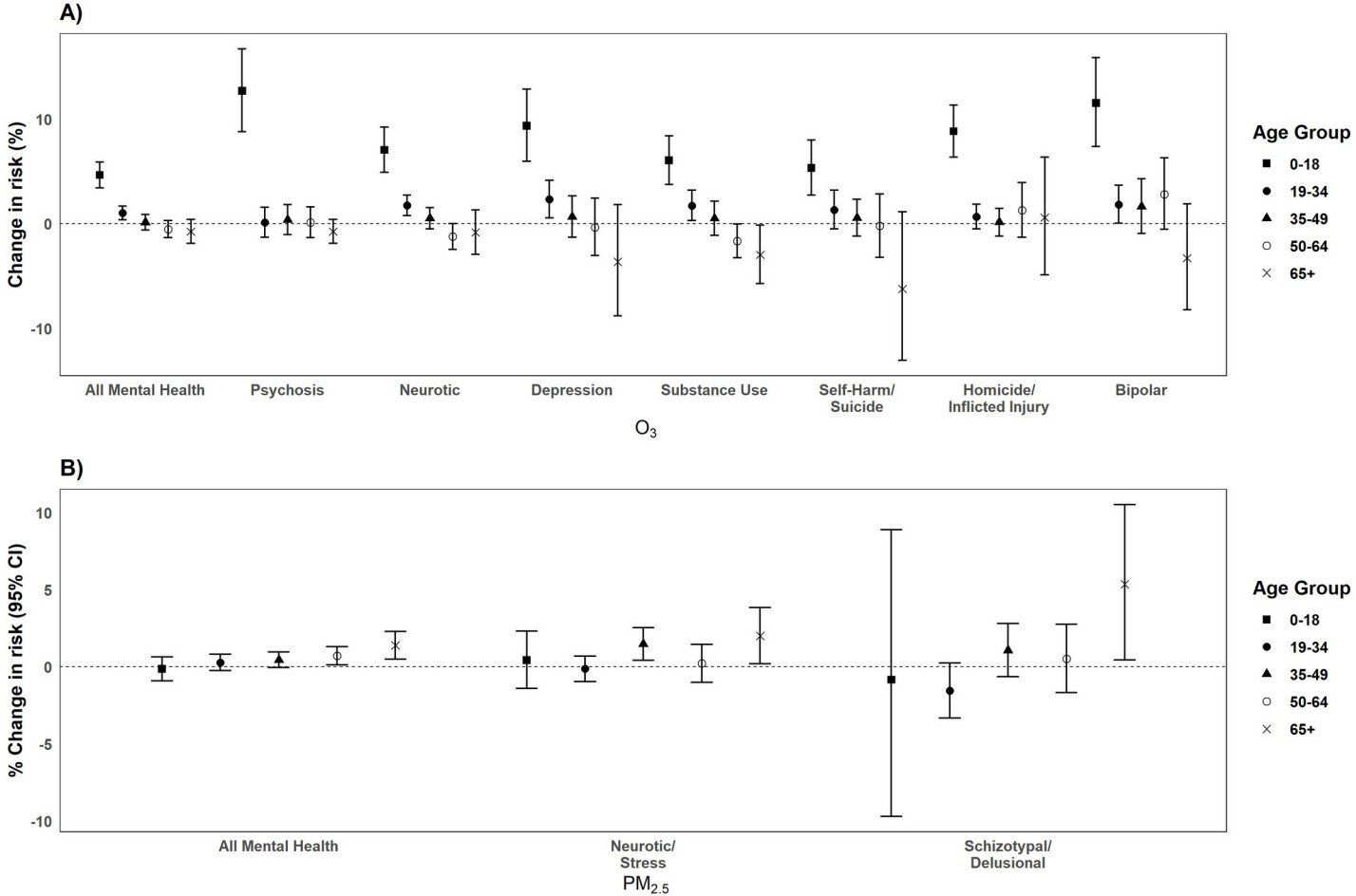

**Fig 6.** Percentage change (95% CI) in the risk of ED visits for $O_3$ (per 10 ppb increase) (A) and $PM_{2.5}$ (per 10μg/m³ increase) by age group from 2005–2013 in California.[1] Bars: 95% Confidence Interval; CI: Confidence Interval; ED: Emergency Department; ppb: parts per billion. [1]Only significant outcomes shown.

3.99, 14.66]. In contrast, $PM_{2.5}$ had significantly increased risks for homicide/inflicted injury [4.45%, 95% CI: 1.22, 7.79] among Asians compared to Whites [0.80%, 95% CI: 0.01, 1.60 ($p_{diff}$ = 0.03)] (Fig 7B). While Hispanics had increased risks for neurotic disorders (0.64%, 95% CI: 0.12, 1.28) and homicide/inflicted injury [1.54%, 95% CI: 0.83, 2.25], these were not significantly different compared to Whites [neurotic: 0.52%, 95% CI: 0.04, 1.0 ($p_{diff}$ = 0.76); homicide/inflicted injury: 0.80%, 95% CI: 0.01, 1.60 ($p_{diff}$ = 0.18)].

### Seasonal variation

After stratifying by warm and cold season, we found evidence of effect modification by season for $O_3$ and ED visits related to all outcomes considered except for neurotic disorders and self-harm/suicide (Fig 8A). All mental health, psychosis, neurotic and stress, mood and affective disorders, depression, bipolar, schizotypal/delusional, and schizophrenia had significant increased changes in risk during the warm season compared to the cold season. During the warm season, the 7-day average of the daily mean level of $O_3$ had a 1.94% increase [95% CI: 0.29, 3.60] in schizotypal/delusional visits, while there was a -3.05% decrease [95% CI: -5.41, -0.65, ($p_{diff}$ = <0.01)] in ED visits during the cold season. Within this broader category, ED visits related to schizophrenia had a 2.33% increase [95% CI: 0.52, 4.16] during the warm season

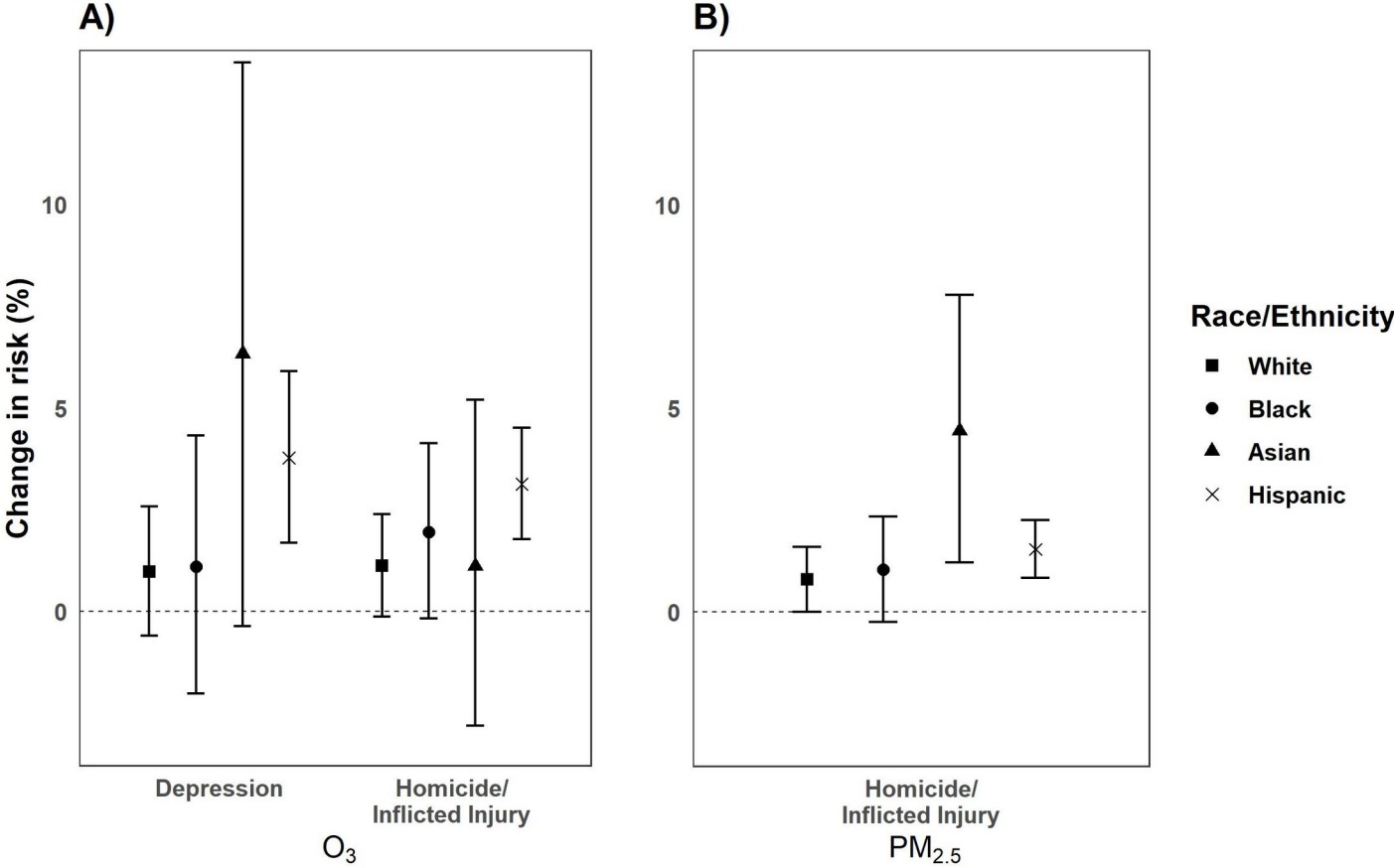

**Fig 7.** Percentage change (95% CI) in the risk of ED visits for $O_3$ (per 10 ppb increase) (A) and $PM_{2.5}$ (per 10μg/m³ increase) (B) by race/ethnicity from 2005–2013 in California.[1] Bars: 95% Confidence Interval; CI: Confidence Interval; ED: Emergency Department; ppb: parts per billion. [1]Only significant outcomes shown.

compared to the cold season [-2.53%, 95% CI: -95% CI -4.72, -0.29 ($p_{diff}$ = <0.01)]. In addition, exposure to $O_3$ was related to a 1.87% increase [95% CI: 0.44, 3.31] for bipolar disorder during the warm season compared to the cold season [-0.69%, 95% CI: -2.64, 1.37 ($p_{diff}$ = 0.05)] (Fig 8A). Homicide/inflicted injury was the only outcome with significantly increased risks during the colder months [2.81%, 95% CI: 1.47, 4.18] compared to the warmer months [-1.67%, 95% CI: -3.04, -0.29, ($p_{diff}$ = <0.01)] for the 30-day average of the daily mean level of $O_3$. We also observed seasonal differences in $PM_{2.5}$ effects with larger associations in the warm season for self-harm/suicide [1.14%, 95% CI: -0.38, 3.69] and homicide/inflicted injury [2.39% increase, 95% CI: 1.45, 3.33] compared to the cold season [self-harm/suicide: -0.71%, 95% CI: -1.69, 0.28 ($p_{diff}$ = 0.05); homicide/inflicted injury: 0.96%, 95% CI: 0.28, 1.64 ($p_{diff}$ = 0.01)] (Fig 8B).

## Socioeconomic-related factors

The distributions of the socioeconomic-related factors were similar for both the $O_3$ and $PM_{2.5}$ study populations. The mean percentages of poverty for $O_3$ and $PM_{2.5}$ air monitor regions were 11 percent and 12.84 percent, respectively. The study's $O_3$ regions had a slightly higher mean percent with a Bachelor's degree (27.1%) compared to $PM_{2.5}$ regions (24.1%). The mean percentages of unemployment for both $O_3$ and $PM_{2.5}$ regions were approximately 7 percent. Median household incomes were, on average, higher in $O_3$ regions ($62,901.09) compared to $PM_{2.5}$ regions ($57,757.58). The mean percentages of obesity for both pollutant study regions

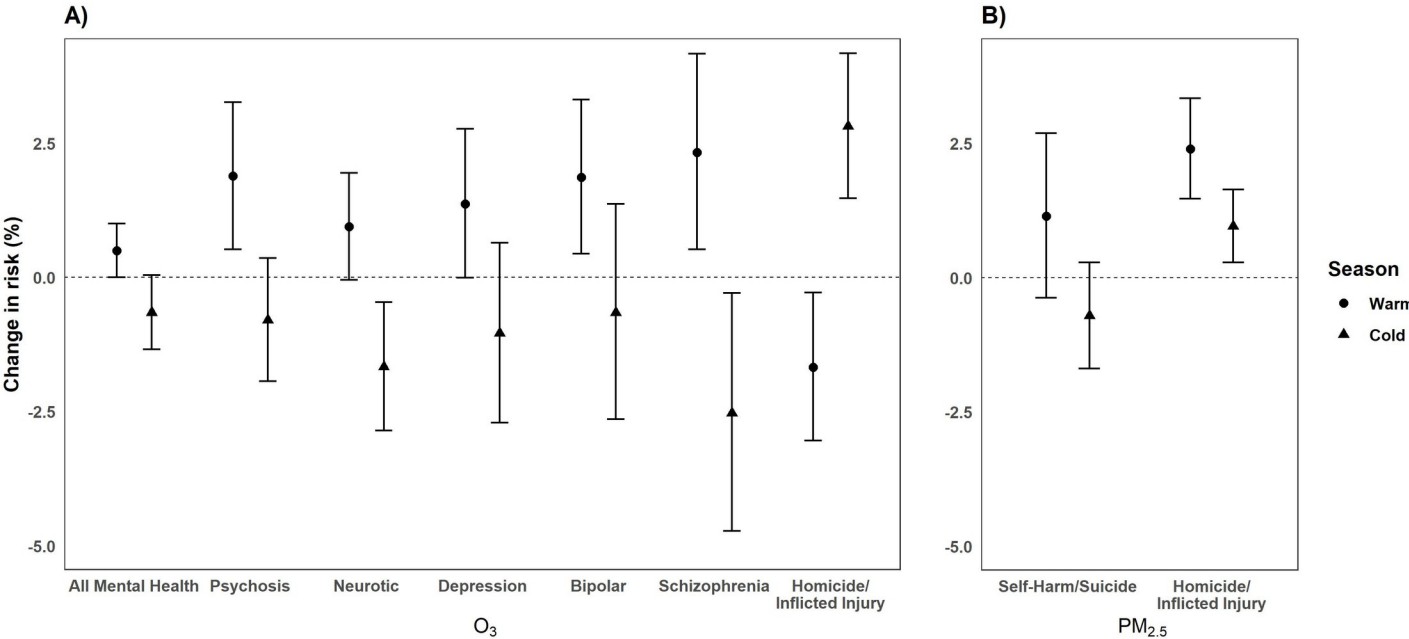

**Fig 8.** Percentage change (95% CI) in the risk of ED visits for $O_3$ (per 10 ppb increase) (A) and $PM_{2.5}$ (per 10μg/m³ increase) (B) by season from 2005–2013 in California.[1] Bars: 95% Confidence Interval; CI: Confidence Interval; Warm: (May-October), Cold: (November-April); ppb: parts per billion. [1]Only significant outcomes shown.

were approximately 25 percent. Associations with $O_3$ and $PM_{2.5}$ appeared to be modified by community SES characteristics for ED visits related to all mental health, neurotic, substance use, mood/affective, self-harm/suicide, and homicide/inflicted injury (results not shown). We did not find significant associations for the other outcomes examined in our study. We found some evidence of increased risk of mental health-related ED visits for $O_3$ in regions with low percent of unemployment and low percent of poverty that differed across mental health outcomes. In regions with low unemployment rates, $O_3$ had significantly increased risks for ED visits related to neurotic disorders [1.82%, 95% CI: 0.99, 2.66], substance use [1.36%, 95% CI: 0.27, 2.47], self-harm/suicide outcomes [2.42%, 95% CI: 0.83, 4.04], and homicide/inflicted injury [2.96%, 95% CI: 1.86, 4.10] compared to regions with high unemployment rate [neurotic: -0.33%, -1.67, 1.03 ($p_{diff} = 0.01$); substance use: -1.46%, 95% CI: -3.14, 0.24 ($p_{diff} = 0.01$); self-harm/suicide: -0.39%, 95% CI: -2.39, 1.65 ($p_{diff} = 0.03$); homicide/inflicted injury: 0.01%, 95% CI: -1.71, 1.77 ($p_{diff} = 0.01$)]. In regions with low rates of poverty, $O_3$ had significantly increased risks for mood and affective disorders [2.24%, 95% CI: 1.12, 3.38] compared to regions with high rates of poverty [-0.63%, 95% CI: -3.19, 2.01 ($p_{diff} = 0.05$)]. We did not observe any findings related to unemployment or poverty for $PM_{2.5}$. Associations with $O_3$ also appeared to be modified by median household income and educational attainment. Higher median income areas showed stronger associations between $O_3$ and substance use-related ED visits [1.0%, 95% CI: -0.12, 2.12] compared to lower median income areas [-0.79%, 95% CI: -2.18, 0.63 ($p_{diff} = 0.05$)]. Regions with higher educational attainment showed stronger associations between $O_3$ and neurotic disorders [2.58%, 95% CI: 0.94, 4.25] compared to regions with lower educational attainment [0.76%, 95% CI: -0.02, 1.55 ($p_{diff} = 0.05$)]. We also did not observe any findings related to income or education for $PM_{2.5}$. Regions with lower obesity rates were linked to stronger relationships between $O_3$ and neurotic [1.92%, 95% CI: 1.08, 2.76], mood/affective [2.26% 95% CI: 1.01, 3.53], substance use [1.46%, 95% CI: 0.36, 2.57],

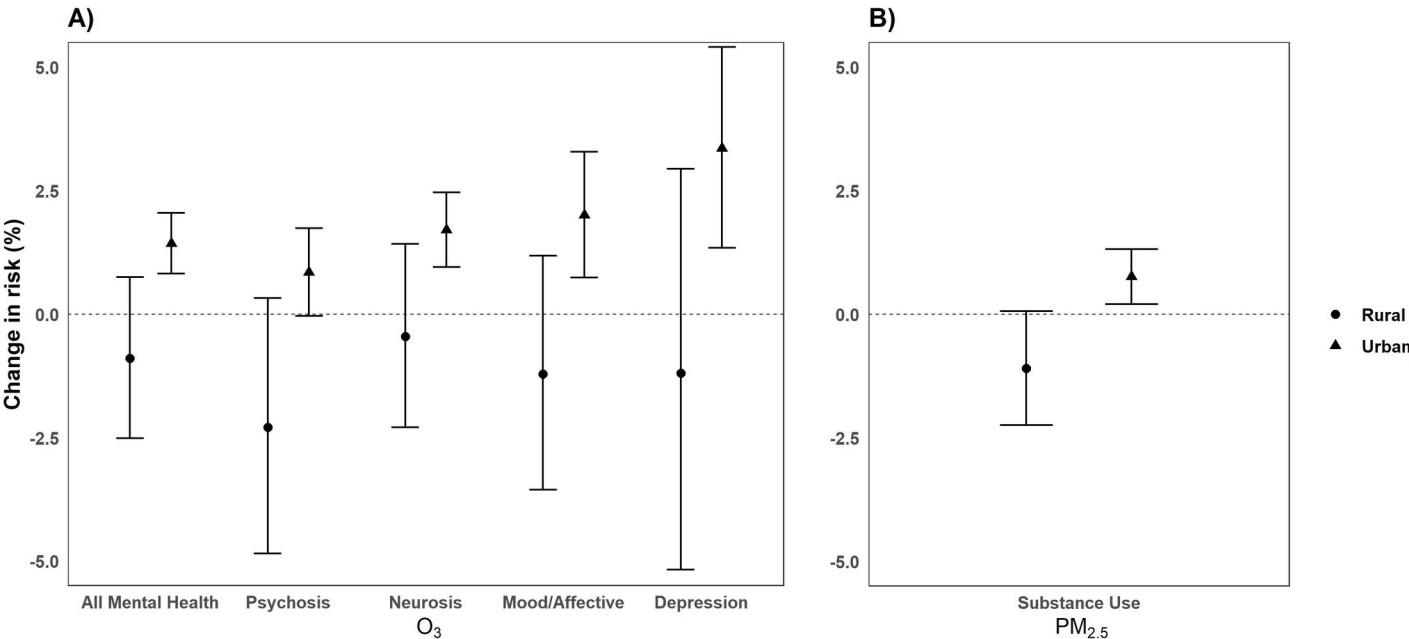

**Fig 9.** Percentage change (95% CI) in the risk of ED visits for $O_3$ (per 10 ppb increase) (A) and $PM_{2.5}$ (per 10μg/m$^3$ increase) (B) by urbanicity from 2005–2013 in California.[1] Bars: 95% Confidence Interval; CI: Confidence Interval; ED: Emergency Department; ppb: parts per billion. [1]Only significant outcomes shown.

self-harm/suicide [2.99%, 95% CI: 1.48, 4.51], and homicide/inflicted injury [3.08%, 95% CI: 1.96, 4.22] compared to regions with higher obesity rates [neurotic: -0.60%, 95% CI: -1.80, 0.61 ($p_{diff} = <0.01$); mood/affective: -0.57%, 95% CI: -2.63, 1.53 ($p_{diff} = 0.02$); substance use: -1.75%, 95% CI: -3.32, -0.15 ($p_{diff} = <0.01$); self-harm/suicide: -1.71%, 95% CI: -3.74, 0.36 ($p_{diff} = <0.01$); homicide/inflicted injury: -0.40, 95% CI: -2.02, 1.24 ($p_{diff} = <0.01$). In contrast, regions with higher obesity rates were linked to stronger association between $PM_{2.5}$ and mood/affective disorder [1.55%, 95 CI: 0.41, 2.71] compared to regions with lower obesity rates [0.18%, 95% CI: -0.44, 0.82 ($p_{diff} = 0.04$)].

**Urbanicity.** We observed significant differences among air monitors located in urban and rural regions for both pollutants (Fig 9). $O_3$ exposure was significantly greater in urban regions compared to rural regions for all mental health, psychosis, neurotic, and mood and affective disorders (Fig 9A). Urban-rural differences for depression were also suggestive of association ($p_{diff} = 0.052$). Mood/affective had the highest percent change in risk in urban regions, with a 2.0% [95% CI: 0.74, 3.28] increase in risk of ED visits compared to the risk in rural regions [-1.22, 95% CI: -3.56, 1.18 $p_{diff} = 0.02$] (Fig 9A). Rural-urban differences for $PM_{2.5}$ were significantly different for substance use (Fig 9B). Same-day exposure to $PM_{2.5}$ was significantly associated with substance use in urban regions [0.76%, 95% CI: 0.21, 1.31] compared to rural regions [-1.09%, 95% CI: -2.24, 0.06 ($p_{diff} = <0.01$)] (Fig 9B).

## Discussion

This study examined short, moderate, and longer-term exposure to ambient $O_3$ and $PM_{2.5}$ on mental health-related ED visits. To our knowledge, this study is the first to assess this association for a wide range of mental health outcomes. We found evidence of associations between exposure to $O_3$ and risk of ED visits related to all mental health, neurotic, mood/affective, depression, bipolar, self-harm/suicide, and homicide/inflicted injuries. We also found evidence of associations between exposure to $PM_{2.5}$ and risk of ED visits related to all mental

health and neurotic disorders. Another time-series study examining the short-term exposures of CO and $NO_2$ on mental health-related ED visits in California was recently published by our group [16]. These two studies together compile the current evidence of air pollutants and mental health outcomes in California.

Previous studies have also found significant associations between air pollutants and several of these mental health outcomes [12, 31–35]. However, our findings on the association between $O_3$ exposure and bipolar disorder is especially novel. Bipolar disorder is understudied in the context of air pollution literature. A recent U.S.-based study found that poor air quality increased the overall risk for bipolar disorder diagnosis. In contrast to our methodology, this study used a summary measure of air quality as their exposure measure and examined long-term mental health effects of air pollutants [36]. We also observed increased risks for homicide/inflicted injury for both $O_3$ and $PM_{2.5}$, which were ED cases for victims of intentional inflicted injuries or homicide. The relationship between exposure to air pollutants and homicide/inflicted injury is poorly understood. A recent U.S. based ecological study examined daily crime activity via the National Incident-Based Reporting System and county-level air quality data [37]. They found that changes in $O_3$ and $PM_{2.5}$ were associated with increased aggravated assaults. One plausible mechanism is that increased levels of air pollutants may impact psychological pathways related to perceived coping, control, and impulsive behavior [37, 38].

We also found evidence of effect modification by season, although findings differed for each pollutant. For $O_3$, we observed significant increased risks for ED visits for the majority of mental health outcomes considered during the warm season and decreased risks during the cold season. Even after controlling for seasonal patterns, $O_3$ exposure had increased risks during the warm season, particularly for schizophrenia and bipolar disorders, both outcomes which share a similar seasonality onset during the warmer months [39–41]. Furthermore, more people may be exposed to ambient air during the warm season when more time is generally spent outdoors. We considered this factor as a potential confounder, thus we controlled for apparent temperature in all of our models.

Findings from previous studies which examined seasonal effects on air pollution and depression are generally consistent. Two studies in Edmonton, Canada found an increased risk between exposure to $O_3$ and depression-related ED visits during the warm season, consistent with findings from our study [14, 15]. One of these studies also found increased risk of depression during the cold season, but this was only observed among females. Overall, $O_3$ had stronger associations during the warm season and this association is in agreement with previous studies [42, 43]. Several studies have also reported increased risks for self-harm/suicide during warmer months following short-term $O_3$ exposure [44–46]. While we observed associations between $O_3$ and ED visits related to self-harm/suicide in our all-year analyses, we did not observe any increased risks during the warm season. Additionally, we found an increased risk for ED visits related to homicide/inflicted injury during the cold season compared to the warm season. Prior research has indicated seasonal variation in homicide and violent events, thus, we hypothesized an increased risk of homicide/inflicted injury visits following $O_3$ exposure during the warm season [23, 47]. However, we observed an association in the opposite direction. In terms of $PM_{2.5}$ exposure, daily mean concentrations are typically higher during the cold season. After controlling for season, we found significantly increased risks for self-harm/suicide visits during the warm season. We also observed significant associations between $PM_{2.5}$ and homicide/inflicted injury visits during the cold season. Season-specific findings for $PM_{2.5}$ are inconclusive. Our findings for self-harm/suicide risk are different from those of previous studies. In contrast to other criteria air pollutants, several studies found increased risks for all mental health, schizophrenia, self-harm/suicide, and depression in colder seasons, particularly for $PM_{2.5}$ and larger fine particles [9, 14, 15, 32, 34, 48]. However, a recent study

found increased risks of self-harm/suicide during both warm and cold season, with no evidence of statistical interaction [31]. One study also found stronger associations between particulate pollutants ($PM_{10}$) and violent suicidal outcomes in colder months but not for nonviolent suicidal outcomes. It is likely that there may be seasonal variations within the broader category of self-harm/suicide that we did not explore.

In general, the prevalence of mental health disorder is higher among females than males, and people aged 18 to 25 years have a higher prevalence of mental health disorders compared to older age groups. In our study, we found that males or females in our study had increased risks of ED visits for different types of mental health outcomes. Several previous studies based in the U.S. and Canada found that females had significantly increased risks for depression with increased air pollutant exposure [14, 15, 49]. While our $O_3$ analysis found that females were at an elevated risk for depression, more than twice the magnitude the risk among males, this was not significantly different than the risk for males, although our study examined short-term exposures while some previous studies focused on long-term exposures. We found significant differences between males and females for substance use in our $O_3$ analyses. $O_3$ exposure was associated with significantly increased risk of substance use-related ED visits among females compared to males. To our knowledge, only one study has examined the relationship between air pollutants and substance use-related ED visits [43]. In this Canadian-based study, the frequency of ED visits was higher among males than females. They examined associations for both $NO_2$ and $PM_{2.5}$ and found that the $NO_2$ association they observed in their main analyses did not significantly differ by sex. We found significant differences between males and females for schizotypal/delusional disorders in our $PM_{2.5}$ analyses. In our $PM_{2.5}$ analyses, females were at increased risk for schizotypal/delusional and schizophrenia disorders, despite the higher frequency of cases ($> 60\%$) among males. Two previous studies examined sex differences in schizophrenia-related ED visits, although findings were discordant [11, 13]. One of these studies found that males were more vulnerable to schizophrenia outcomes compared to females, and this was likely due to a greater proportion of males in that study population having outdoor occupations [11]. Findings from studies conducted in smaller regions are not generalizable to our study population.

Compared to other age groups, those aged 0–18 years were at greatest risk from $O_3$ for some of our mental health outcomes, except for schizophrenia and schizotypal/delusional disorders, which generally manifest during adulthood and are rare among adolescents [50]. There is growing research to support the relationship between air pollutants and several mental health outcomes among adolescents. A recent U.K.-based study found that adolescents exposed to higher concentrations of outdoor air pollutants were more likely to report psychotic experiences/symptoms (e.g., paranoia, hearing voices) compared to those exposed to lower levels of air pollutants [51]. While there are limited findings for the other mental health outcomes examined in our study, risk of increased psychotic disorders among adolescents is of particular importance partly because adolescents who experience psychotic symptoms are at an increased risk of other mental health conditions such as suicidal behavior or bipolar disorder [52, 53]. Another recent study based in Cincinnati, Ohio found that short-term exposure to $PM_{2.5}$ was associated with increased risks of psychiatric emergency department visits related to schizophrenia, suicidality, adjustment, and anxiety disorders among adolescents [54]. The study found stronger associations for suicidality and anxiety-related ED visits among children who lived in socioeconomically deprived neighborhoods, which suggests that chronic exposure to neighborhood-level stressors may modify the relationship between air pollutants and mental health among children. These recent findings warrant future studies to examine these relationships for other criteria air pollutants and other populations.

Older age groups, particularly elderly populations, are more susceptible to depressive symptoms. Some research indicates that increased concentration levels of air pollutants, including $NO_2$, $O_3$, and $PM_{10}$, are associated with increased risk of depressive symptoms among elderly populations [49, 55]. We did not observe any increased risks for depression or any other mental health conditions among older adults in our $O_3$ analyses. However, we found increased risks for schizotypal/delusional and neurotic and stress ED visits among older populations following $PM_{2.5}$ exposure. The association between air pollutants and schizotypal/delusional-related outcomes among older populations have rarely been examined. Though our findings are in agreement with a recent Japanese-based study. The study found strong associations between increased levels of $PM_{2.5}$ exposure and schizophrenia severity, particularly among patients aged 65 and older. A general explanation for these findings may be related to the effects of aging, in that older populations are more vulnerable to $PM_{2.5}$ exposure. However, this was only apparent in two of our mental health outcomes. Findings from several studies suggest that air pollutants may worsen depressive symptoms among those 65 years and older [55, 56]. Like a lot of the mental health outcomes examined, there are overlapping symptoms between schizophrenia and depression. Individuals with schizophrenia are likely to be diagnosed with depression or experience depressive symptoms and schizophrenia and depression comorbidity is common among older adults [57, 58]. Our outcome data is most likely capturing late-onset schizophrenia or cases of relapse among this population. Our lack of findings on depression among elderly group may also be due to examining ED visit data that only represented severe and urgent cases of depression. Previous studies examined air pollution and depression among elderly populations via self-reported questionnaires, medical visits, or prescription medication use data, focusing on chronic exposure, though these studies resulted in inconsistent findings [42, 49, 59]. While elderly populations are at increased risk for major mental health disorders, they may not have as much exposure to ambient air pollutant as other age groups. Survey results from a multi-city U.S. based study showed that older adults between the ages 45–84 years spend roughly 72% of their time indoors at home [60]. Furthermore, previous studies have suggested that elderly populations with diagnosed mental health conditions are also more likely to stay indoors if residing in an assisted living facility [23, 60, 61].

All race/ethnicity groups examined in our study were at increased risk for at least one mental health-related ED visit. However, Asians and Hispanics had significantly greater risks compared to Whites for depression and homicide/inflicted injury. To date, no studies on air pollutants and mental health have examined differences among race/ethnicity groups for $PM_{2.5}$ and $O_3$ specifically. Studies have only included race/ethnicity as a confounder in their models. Other studies which used hospital or ER visit data state limitations in obtaining individual-level data on race/ethnicity. Though research findings specific to ground-level $O_3$ are limited, a recent study found that Asians are exposed to higher concentrations of other air criteria pollutants compared to Whites [62]. Similar to a previous study examining mental health-related ED visit data, small sample size for ED visit data is a concern [23]. There were few Asians and Blacks cases in our study, thus our estimates may not accurately reflect the racial/ethnic disparities for mental health-related ED visits.

While air quality poses as a health risk for everyone, those who live in socioeconomically disadvantaged communities are more likely to be exposed to higher concentrations of criteria air pollutants [3]. We used aggregated data on median household income, proportion below the federal poverty threshold, employment rates, and educational attainment as proxies for socioeconomic (SES). for each air monitor region. We hypothesized that we would observe increased risks of ED visits in socioeconomically deprived regions as well as in regions with high percent of obesity. Our hypothesis is supported by results from a meta-analysis which found that employment and occupation status affected vulnerability to $O_3$ exposure [63].

However, findings from our $O_3$ analysis did not confirm our hypothesis. We found evidence of effect modification in regions with low percent of unemployment, low percent obesity, low percent of poverty, and high median household income. Another study based in Australia found that areas with greater socioeconomic disadvantage were exposed to higher concentrations of $PM_{2.5}$ [64]. In our $PM_{2.5}$ analysis, regions with higher rates of obesity had significantly increased risks for mood and affective disorders. While obesity has been linked with mood disorders, our aggregated data did not allow us to make such inferences from our results [65]. There are several possible explanations for our findings. First, pollutant concentrations were similar in our study's low SES communities and high SES communities. Second, our inconsistent findings may be due to the structure of our data and operationalization of SES. When we stratified our data, we found that a low proportion of the study air monitors were located in regions with lower SES, which may have skewed our findings. Furthermore, SES is a complex construct and may not be fully captured by the variables we examined. While SES is a risk factor for mental health, this relationship varies across disorder types and is further complicated by individual-level characteristics such as genetic predisposition, sex, and race/ethnicity [66]. Our limited findings in understanding the impact of SES on air pollution and mental health underscores the need for future studies to elucidate this relationship.

Geographical variation may pose as a major risk factor for many mental health disorders, including neurotic, mood and affective disorders, psychoses, and substance use. Previous studies show that the rates of major mental health disorders are higher in urban areas compared to rural areas, due to factors related to overcrowding and physical neighborhood stressors [2, 67]. To our knowledge, very few studies have accounted for urban-rural differences in examining the relationship between air pollutants and mental health. Using population density as a proxy measure for urban-rural differences, we found that the effects of $O_3$ exposure on neurotic, mood and affective, and depression disorders and $PM_{2.5}$ exposure on substance use disorders were significantly higher in urban regions compared to rural regions. Exposure to air pollutants are also greater in urban areas where there are higher concentrations of industrial exhaust and vehicle emissions via traffic [2]. Our urban-rural results may be skewed since most air monitors were located in more densely populated regions to represent exposures for the majority of the population and help with monitoring and regulating criteria air pollutants. Thus, our study's "rural" areas may be accurately described as semi-rural. In addition, population density is a limited measure for urbanicity since it is determined by arbitrarily defined areas of land [68].

Chronic stressful conditions, life events, and physiological factors can all impact psychological health. The biological mechanisms through which exposure to pollutants affects mental health are complex and not well understood [9]. Inhaling harmful air pollutants can damage the cardiovascular and respiratory systems through inflammation, microglial activation, and oxidative stress mechanisms. These mechanisms may impact the central nervous system in similar ways leading to neurological and psychiatric disorders [69–71]. One plausible pathway is chronic inflammation of the upper and lower respiratory tracts which leads to a systemic inflammatory response, producing harmful inflammatory mediators and reactive oxygen species in the central nervous system. An imbalanced production will eventually result in oxidative stress, causing the human brain to become susceptible to neurodegeneration and neuroinflammation, processes associated with the development and exacerbation of mental disorders [70]. These proposed mechanisms are supported by findings from several recent animal models. Brain tissue retrieved from rodents exposed to ambient fine particles exhibited up-regulation of genes in inflammatory cytokine pathways and inflammation to the hippocampus [36, 72]. Findings from another animal study also showed that $PM_{2.5}$ inhalation resulted in changes in mitochondrial structure and function which may then lead to cognitive

impairment and neurodegeneration [8]. As a reactive oxygen species, acute or chronic $O_3$ exposure can cause neuroinflammation through increased cytokine production in the brain, braid lipid peroxidation, and neuron damage [7]. While the exact mechanisms are not fully understood, findings from animal studies supports that exposure to air pollution stimulates systemic oxidative stress, neuroinflammation, and neurodegenerative pathways [6, 7, 69].

Our study was subject to several limitations. We were not able to identify cases who had multiple visits to the ED during the study period. We also were not able to examine co-occurring mental health outcomes, which may be common among individuals with substance use disorders [73]. Our study data did not include individual-level characteristics related to SES, family medical history, medication use, and/or access to mental health care services or other time-varying confounders such as traffic or other meteorological factors. These variables, among others, are possible moderators influencing the relationship between air pollutants and mental health [74]. In addition, some of our outcome categories had small sample sizes, especially when stratified by sub-population. We were also not able to further explore specific mental health-related outcomes in detail, such as self-harm/suicide and homicide/inflicted injury. Cases related to homicide/inflicted injury refer to victims of violence, not the individuals inflicting the violence. In our analysis, we assume the perpetuators of homicide/inflicted injury-related cases had the same air pollutant exposure experiences as the victims themselves. Our study results may not be generalizable to other regions and/or may not be fully representative of California, especially since air monitors are typically located in semi-urban and urban regions.

Nonetheless, we examined a wide range of mental health outcomes, including several that are understudied outcomes, such as homicide/inflicted injury and bipolar disorder. Previous studies focused on 'all mental health-related' outcomes or broader sub-categories of mental health outcomes. In addition, we were able to identify potential vulnerable populations by examining outcomes by race/ethnicity, sex, age group, SES, and urbanicity, addressing critical knowledge gaps in the current literature [9]. Our study also used data covering a larger geographic area of California than most existing studies which examined air pollutant data in single cities. In this approach, our pooled estimates across statewide air monitors are more likely to be robust and less vulnerable to bias [75]. We conducted our model building, selection, and sensitivity analyses using several approaches as recommended in previous studies [9, 10, 74]. These approaches included controlling for major time-variant factors (seasonality, temperature), modeling alternative long-term trends, and examining different structures and lengths in lags.

Harmful levels of air pollutants remain a significant concern worldwide. In addition to established physical health impacts related to cardiovascular and respiratory illnesses, there is increasing evidence supporting the effects of air quality on mental health. Our study contributes to the limited U.S.-based literature examining the relationship between short-term $O_3$ exposure and mental health. In addition to our main findings, we identified several vulnerable populations susceptible to mental health impacts of $O_3$ and $PM_{2.5}$ exposure, including women, those 18 years and younger, and specific racial/ethnic groups depending on the outcome. Considering these results, further investigation of vulnerable populations across different mental health conditions is warranted. Our results support the need for policy makers to develop targeted prevention efforts to protect vulnerable groups from being adversely affected by air pollution.

## Conclusions

This study found evidence of a positive association between exposure to $O_3$ and $PM_{2.5}$ and all mental-health related ED visits in California. Furthermore, we found evidence of increased

risks for sub-categories of mental health including neurotic, mood and affective, depression, bipolar, self-harm/suicide, and homicide/inflicted injury outcomes. The effects of $O_3$ and $PM_{2.5}$ were also stronger during warmer months. We also found that adolescents, elderly populations, females, Asians, and Hispanics were particularly vulnerable to the effects of pollutant exposure and this varied by mental health outcomes. While the research literature is increasing, there still exists a knowledge gap on how air pollution affects mental health among subgroups. Future epidemiological studies should continue examining the effects of pollutant exposure by gender, age, and race/ethnicity. Our study's findings also support the need to better understand the biological mechanisms between exposure to air pollution and mental health outcomes.

## Acknowledgments

Author affiliations: School of Public Health, University of California—Berkeley, (Angela-Maithy Nguyen, MPH) and Air and Climate Epidemiology Section, California Office of Environmental Health Hazard Assessment, Oakland, California (Rupa Basu, PhD, MPH, Brian Malig, MPH). We thank Keita Ebisu for their methodological guidance. We also thank Ruwan Thilakaratne and Rachel Broadwin for reviewing our manuscript and providing thoughtful feedback. The opinions expressed in this article are those of the authors and do not represent those of the California Environmental Protection Agency or the Office of Environmental Health Hazard Assessment.

## Author Contributions

**Conceptualization:** Brian J. Malig, Rupa Basu.

**Data curation:** Angela-Maithy Nguyen, Brian J. Malig.

**Formal analysis:** Angela-Maithy Nguyen.

**Methodology:** Brian J. Malig, Rupa Basu.

**Software:** Angela-Maithy Nguyen, Brian J. Malig.

**Supervision:** Rupa Basu.

**Visualization:** Angela-Maithy Nguyen.

**Writing – original draft:** Angela-Maithy Nguyen.

**Writing – review & editing:** Brian J. Malig, Rupa Basu.

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
