## [Decision Letter · Decision Letter 0]

7 Jan 2021

PONE-D-20-33930

The association between ozone and fine particles and mental health-related emergency department visits in California, 2005-2013

PLOS ONE

Dear Dr. Nguyen,

Thank you for submitting your manuscript to PLOS ONE. After careful consideration, we feel that it has merit but does not fully meet PLOS ONE’s publication criteria as it currently stands. Therefore, we invite you to submit a revised version of the manuscript that addresses the points raised during the review process.

We look forward to receiving your revised manuscript.

Kind regards,

Xiaohui Xu, PhD

Academic Editor

PLOS ONE

Journal Requirements:

Reviewers' comments:

Reviewer's Responses to Questions

**Comments to the Author**

1. Is the manuscript technically sound, and do the data support the conclusions?

Reviewer #1: Partly

Reviewer #2: Partly

2. Has the statistical analysis been performed appropriately and rigorously? 

Reviewer #1: No

Reviewer #2: No

3. Have the authors made all data underlying the findings in their manuscript fully available?

Reviewer #1: No

Reviewer #2: Yes

4. Is the manuscript presented in an intelligible fashion and written in standard English?

Reviewer #1: Yes

Reviewer #2: Yes

5. Review Comments to the Author

Reviewer #1: Nguyen et al. examined the associations between air pollutants and emergency department (ED) visits related to mental health disorders in California between 2005 and 2013. The authors found that ambient ozone and PM2.5 are associated with higher risks of ED visits due to certain mental health disorders. However, several aspects of outcome ascertainment, statistical analysis, and discussion need more work. The manuscript could be suitable for publication with major revisions.

Major comments.

1. Outcome data.

(1) Please describe the number of air pollutant monitors, their locations, and the area of each monitoring site. It would be helpful to add a table to include that information.

(2) Please explain why ICD-9-CM codes, instead of ICD-10-CM, were utilized to identify mental health outcomes. The recent adoption of ICD-10-CM coding in the United States occurred nationwide in 2015. ICD-10-CM represents a major revision, designed to better support the role of coding in reimbursement, quality measurement, and monitoring.

2. Statistical analysis.

(1) Was a case-crossover design implemented to account for potential confounders such as age, sex, and socioeconomic status? If not, please describe how confounding was considered in the analysis.

(2) page 12. lines 113-114. The description of statistical models is unclear. Please describe how daily temperature, relative humidity, and wind speed, and their non-linear and lagged effects were taken into account in the analysis.

3. Results.

(1) Table 1. Please include the descriptive statistics of socioeconomic status and all other covariates in Table 1.

(2) Please report all the study results for PM2.5 and ozone at lag0, lag1, lag2, lag6, lag0-6, and lag0-29 for all the analyses. It is inappropriate to report only part of the results.

(3) Please include p values for interactions in Figures 3-7.

4. Discussion. It is imperative to summarize biological evidence to support the observed associations.

Minor comments.

1. Abstract. Page 8, lines 35-37. Please consider revising the last sentence. What are the public health implications of the findings?

2. Introduction. Page 9. lines 62-64. What is the study hypothesis? It is better to discuss the study results in Results and Discussion.

Reviewer #2: This manuscript evaluated the associations between ozone and fine particles and mental health-related emergency department visits in California. This work contributed to the limited evidence concerning the adverse effects of air pollutants on mental disorders. I have some major comments to improve this analysis. Maybe reanalysis is needed.

1. In the introduction section, please provide some rationales why focused on PM2.5 and ozone, rather than other air pollutants.

2. In the statistical analysis section, only the same-day mean apparent temperature was adjusted. It is well known that the health effects of temperature can last for weeks.

3. In the statistical analysis section, it is not clear whether the nonlinear effects of temperature were adjusted. Please show more details.

4. In the statistical analysis section, the authors controlled for seasonal or long-term trend by using a natural cubic spline measured with 2 degrees of freedom per year. Only 2 degrees of freedom per year were apparently not enough. Typically, at least 7 or 8 df is needed.

5. In the statistical analysis section, why the authors examined the effects of PM2.5 used only one-day lag? In contrast, the lag to be evaluated is weeks or months for ozone.

6. A section of limitations is missing.

6. PLOS authors have the option to publish the peer review history of their article (what does this mean?). If published, this will include your full peer review and any attached files.

Reviewer #1: No

Reviewer #2: No

---

## [Author Response · Author response to Decision Letter 0]

22 Feb 2021

To: Reviewers of PONE-D-20-33930

We would like to thank you for your time and effort in reviewing our manuscript. The suggestions and feedback we received were insightful and helpful during the revision process. In addition to revising the manuscript, we have provided responses to each of your comments below. 

Response to Reviewers

Reviewer #1: Nguyen et al. examined the associations between air pollutants and emergency department (ED) visits related to mental health disorders in California between 2005 and 2013. The authors found that ambient ozone and PM2.5 are associated with higher risks of ED visits due to certain mental health disorders. However, several aspects of outcome ascertainment, statistical analysis, and discussion need more work. The manuscript could be suitable for publication with major revisions.

Major comments.

1. Outcome data.

(1) Please describe the number of air pollutant monitors, their locations, and the area of each monitoring site. It would be helpful to add a table to include that information.

We briefly described the area of each monitoring site in our Methods section (line 115-118), the number of air pollutant monitors in our Descriptive findings section of the Results (lines 197-198; 218). We also included two maps showing the location sites of the O3 and PM2.5 air pollutant monitors in our study (Figures 1 and 2).

(2) Please explain why ICD-9-CM codes, instead of ICD-10-CM, were utilized to identify mental health outcomes. The recent adoption of ICD-10-CM coding in the United States occurred nationwide in 2015. ICD-10-CM represents a major revision, designed to better support the role of coding in reimbursement, quality measurement, and monitoring.

We utilized ICD-9-CM codes because our analysis covered the years 2005 through 2013. ICD-9-CM codes capture emergency department visits prior to 2015. 

2. Statistical analysis.

(1) Was a case-crossover design implemented to account for potential confounders such as age, sex, and socioeconomic status? If not, please describe how confounding was considered in the analysis.

No, a case-crossover designed was not implemented to account for confounding. Data on age, sex, and socioeconomic status were only available as count data (e.g., number of male patients to emergency department visits on a given day). We assessed these factors as potential effect modifiers in our stratified analyses. We considered daily mean apparent temperature, national holidays, and seasonal/long-term trends as confounders, which we included in all of our models as covariates (revised lines 121-122). Since our study was a time series design, we were concerned with time-varying covariates, thus including a trend term would control for longer term demographic changes.

(2) page 12. lines 113-114. The description of statistical models is unclear. Please describe how daily temperature, relative humidity, and wind speed, and their non-linear and lagged effects were taken into account in the analysis.

We took into account daily mean apparent temperature, a heat index which incorporates temperature and relative humidity in its calculation (revised lines 97-98). This method was documented in several cited studies (Basu et al., 2008; Basu et al., 2017). Mean apparent temperature was adjusted for in all of our statistical models. We chose to use same-day lag for mean apparent temperature in order to keep our models as parsimonious as possible. In addition, same-day or short-term mean temperature are more likely to be correlated with short-term pollution, thus more likely to confound the pollution effects. Results from a study on temperature and mental health showed that stronger effects were observed for short-term temperature (added lines 124-128). We did not include wind speed in our statistical models.

3. Results.

(1) Table 1. Please include the descriptive statistics of socioeconomic status and all other covariates in Table 1.

We added a brief description of the socioeconomic-related factors (lines 379-386). We did not include this summary in Table 1 since these variables were aggregate-level data we used for our secondary analyses. Gender, race/ethnicity, and age group were count data cases of emergency department visits, and the distribution of these cases are shown in Table 1. 

(2) Please report all the study results for PM2.5 and ozone at lag0, lag1, lag2, lag6, lag0-6, and lag0-29 for all the analyses. It is inappropriate to report only part of the results.

We incorporated this suggestion and included a new table (Table 3) which presents the results for single day lags 0, 1, 2, 7, 0-6, and 0-29 for both pollutants and all mental health-related outcomes. We did not explore cumulative lags 0-6 or 0-29 for PM2.5 due to the proportion of missing data. We edited the text to clarify this (lines 138-143).

(3) Please include p values for interactions in Figures 3-7.

We chose not to include p-values in Figures 3-7 (now Figures 5-9) because they would be hard to see in the plots. However, we highlighted statistically significant interac4ions with their associated p-values in the Results sections (under Sub-group analyses, Seasonal variation, and Socioeconomic-related factors). We determined statistical significance for interaction at the p<.05 level (lines 147-148).

4. Discussion. It is imperative to summarize biological evidence to support the observed associations.

A summary of biological mechanisms and plausible pathways are discussed in lines 619-639. We also added some additional references and details on biological mechanisms in lines 633-637.

Minor comments.

1. Abstract. Page 8, lines 35-37. Please consider revising the last sentence. What are the public health implications of the findings?

We added public health implications in lines 40-41. These implications are that our study’s findings warrant further investigation to better understand how exposure to air pollutants impact vulnerable groups. We also made minor edits to the Abstract as to not exceed 300 words.

2. Introduction. Page 9. lines 62-64. What is the study hypothesis? It is better to discuss the study results in Results and Discussion.

We included our main study hypothesis in the Introduction (lines 62-68). We removed the study results lines as suggested.

Reviewer #2: This manuscript evaluated the associations between ozone and fine particles and mental health-related emergency department visits in California. This work contributed to the limited evidence concerning the adverse effects of air pollutants on mental disorders. I have some major comments to improve this analysis. Maybe reanalysis is needed.

1. In the introduction section, please provide some rationales why focused on PM2.5 and ozone, rather than other air pollutants.

We focused our analysis on O3 and PM2.5 because of the increasing literature indicating that these pollutants are linked to oxidative stress and neuroinflammation, which we discuss in the Discussion section (lines 622-642). In addition, a recent study published by our group (OEHHA) examined the association between short-term exposure to CO and NO2 and mental health ED visits. We referred to this study in the Introduction and Discussion sections. We added these rationales in lines 49-51 and 62-68 in the Introduction.

2. In the statistical analysis section, only the same-day mean apparent temperature was adjusted. It is well known that the health effects of temperature can last for weeks.

We chose to use same-day mean apparent temperatures because same-day or short-term mean apparent temperature is more likely to be correlated with short-term air pollution, thus more likely to confound the pollution effects. In a previous study on ambient temperature and mental health-related emergency department visits, results showed that stronger effects were observed for short-term temperature. In addition, we adjusted for seasonal/long-term trends which would adjust for some of the longer-term relationships between apparent temperature and mental health outcomes (added lines 124-128).

3. In the statistical analysis section, it is not clear whether the nonlinear effects of temperature were adjusted. Please show more details.

We did not adjust for the nonlinear effects of temperature. Effect estimates from a previous study on ambient temperature and mental health-related emergency department visits showed that same-day lag for mean apparent temperature was the most significant and best fit relative to other lags for that analysis.

4. In the statistical analysis section, the authors controlled for seasonal or long-term trend by using a natural cubic spline measured with 2 degrees of freedom per year. Only 2 degrees of freedom per year were apparently not enough. Typically, at least 7 or 8 df is needed.

In our exploratory analyses, we compared controlling for seasonal/long term trends using 2 degrees of freedom (df) versus 3 and 4 df. The associations were robust to the additional df for time trends in the models. We also found that the qAIC in our models worsened for both 3 df and 4 df, thus we decided to use 2 df. We added this explanation in in lines 134-136. While we acknowledge several studies having using 6-8 df, previous studies focused on California (Thilakaratne et al., 2020; Basu et al., 2017) have used 2-4 df. These df have typically been found to be the best model fit to adjust for seasonal trends. In addition, we reference an article by Bhaskaran et al., 2013, who present the general features and guidelines in conducting time series regression analyses in environmental epidemiology. In this article, they state that too many df may “result in a very ‘wobbly’ function which may compete with the variable of interest to explain the short-term variation of interest, widening confidence intervals of relative risk estimates.” 

5. In the statistical analysis section, why the authors examined the effects of PM2.5 used only one-day lag? In contrast, the lag to be evaluated is weeks or months for ozone.

We explored single day lags up to 7 days and cumulative 1-day lags for PM2.5. We added a new table which shows PM2.5 results for lags 0, 1, 2, and 7 for all mental health outcomes (Table 3). We did not explore cumulative lags beyond 1 day due to the proportion of missing data for PM2.5 There was a large proportion of missing data due because PM2.5 measurement was typically collected every third or sixth day. We have edited the text to clarify this (lines 139-143).

6. A section of limitations is missing.

Our section of limitations is in the Discussion section, lines 643-657.

---

## [Decision Letter · Decision Letter 1]

23 Mar 2021

The association between ozone and fine particles and mental health-related emergency department visits in California, 2005-2013

PONE-D-20-33930R1

Dear Dr. Basu,

We’re pleased to inform you that your manuscript has been judged scientifically suitable for publication and will be formally accepted for publication once it meets all outstanding technical requirements.

Kind regards,

Xiaohui Xu, PhD

Academic Editor

PLOS ONE

Additional Editor Comments (optional):

Reviewers' comments:

Reviewer's Responses to Questions

**Comments to the Author**

1. If the authors have adequately addressed your comments raised in a previous round of review and you feel that this manuscript is now acceptable for publication, you may indicate that here to bypass the “Comments to the Author” section, enter your conflict of interest statement in the “Confidential to Editor” section, and submit your "Accept" recommendation.

Reviewer #1: All comments have been addressed

Reviewer #2: All comments have been addressed

2. Is the manuscript technically sound, and do the data support the conclusions?

Reviewer #1: Yes

Reviewer #2: Yes

3. Has the statistical analysis been performed appropriately and rigorously? 

Reviewer #1: Yes

Reviewer #2: Yes

4. Have the authors made all data underlying the findings in their manuscript fully available?

Reviewer #1: No

Reviewer #2: No

5. Is the manuscript presented in an intelligible fashion and written in standard English?

Reviewer #1: Yes

Reviewer #2: Yes

6. Review Comments to the Author

Reviewer #1: (No Response)

Reviewer #2: The authors have adequately addressed my concerns and I have no further comments. This work is valuable for preventing mental disorders due to air pollution.

7. PLOS authors have the option to publish the peer review history of their article (what does this mean?). If published, this will include your full peer review and any attached files.

Reviewer #1: No

Reviewer #2: No

---

## [Editor Report · Acceptance letter]

25 Mar 2021

PONE-D-20-33930R1 

The association between ozone and fine particles and mental health-related emergency department visits in California, 2005-2013 

Dear Dr. Basu:

I'm pleased to inform you that your manuscript has been deemed suitable for publication in PLOS ONE. Congratulations! Your manuscript is now with our production department. 

Kind regards, 

on behalf of

Dr. Xiaohui Xu 

Academic Editor

PLOS ONE